# Trait anxiety is associated with hidden state inference during aversive reversal learning

Ondrej Zika [1,2] ✉, Katja Wiech [3], Andrea Reinecke [4,5], Michael Browning [4,5] & Nicolas W. Schuck [1,2,6] ✉

Updating beliefs in changing environments can be driven by gradually adapting expectations or by relying on inferred hidden states (i.e. contexts), and changes therein. Previous work suggests that increased reliance on context could underly fear relapse phenomena that hinder clinical treatment of anxiety disorders. We test whether trait anxiety variations in a healthy population influence how much individuals rely on hidden-state inference. In a Pavlovian learning task, participants observed cues that predicted an upcoming electrical shock with repeatedly changing probability, and were asked to provide expectancy ratings on every trial. We show that trait anxiety is associated with steeper expectation switches after contingency reversals and reduced oddball learning. Furthermore, trait anxiety is related to better fit of a state inference, compared to a gradual learning, model when contingency changes are large. Our findings support previous work suggesting hidden-state inference as a mechanism behind anxiety-related to fear relapse phenomena.

Aversive memories are notoriously difficult to forget, and often resist attempts to overwrite them with new experiences. In exposure therapy, for instance, the feared situation or object is presented in the absence of an aversive outcome to achieve extinction of the fear response. While this procedure can lead to a decrease in fear responding, this reduction sometimes remains specific to the therapeutic context, and fails to generalize to the outside world[1,2]. Such deficits in updating of aversive beliefs have been linked to anxiety disorders. Clinical anxiety has been associated with lowered discrimination between conditioned and unconditioned stimuli[3,4], decreased inhibition of responses to conditioned stimuli[5,6] and heightened fear generalization[7]. Some research has suggested that, even in healthy adults, heightened trait anxiety can lead to overly context-specific learning, as indicated by lower success of cognitive behavioral therapy in high trait anxious individuals[8], suboptimal uncertainty adjustment of learning in volatile environments[9,10] and higher rates of fear relapse following treatment[11,12].

One important aspect of aversive learning is the degree to which individuals internally represent periods of relative safety and threat as separate contexts[13], such as acquisition and extinction phases in conditioning studies. The term context can either be used to describe directly perceivable aspects of an environment, such as different virtual rooms[14], or un-cued aspects that have to be inferred based on changing outcomes. The present paper is concerned with the distinction between periods of high or low threat, which represents the latter type of context. We refer to this context as 'hidden state', akin to the idea of partially observable states as discussed in the reinforcement learning (RL) literature[15]. Previous theoretical[16] and experimental[17–19] work suggests that the brain uses such hidden states to represent information and to drive decisions (see ref. 20 for a review).

Learning in a state-dependent manner[21,22] is often contrasted with gradual learning as proposed by classical associative learning theories[22]. The key distinction between the two ways of learning is that, under gradual learning, the individual updates their expectation on a trial-by-trial basis, effectively overwriting their previous estimate with each update. On the other hand, an agent learning in a state-dependent manner creates classes of similar experiences (i.e., states) and

[1]Max Planck Research Group NeuroCode, Max Planck Institute for Human Development, Berlin, Germany. [2]Max Planck UCL Centre for Computational Psychiatry and Aging Research, Berlin, Germany. [3]Wellcome Centre for Integrative Neuroimaging (WIN), Nuffield Department of Clinical Neurosciences, University of Oxford, Oxford, UK. [4]Department of Psychiatry, University of Oxford, Oxford, UK. [5]Oxford Health NHS Trust, Warneford Hospital, Oxford, UK. [6]Institute of Psychology, Universität Hamburg, Hamburg, Germany. ✉e-mail: zika@mpib-berlin.mpg.de; schuck@mpib-berlin.mpg.de

incorporates this information into their predictions of future outcomes. Consequently, state-dependent learning often leads to abrupt jumps in an agent's predictions, called state switches, which reflect when a state change was detected and implemented into the next prediction. In addition, such a learning mechanism can also explain a persistence of previous experiences despite new learning, similar to relapse phenomena observed in clinical practice during which clinically extinguished fear fails to generalize to everyday life[1].

Experiments that aim to arbitrate between these perspectives typically use extinction learning designs, in which a neutral stimulus is first paired with an aversive unconditioned stimulus (US, typically a painful shock) during an acquisition phase, but is no longer followed by shocks during an extinction phase. The gradual learning perspective assumes that shock contingencies during the acquisition phase lead to gradual strengthening of a cue-outcome association, which is then gradually weakened, and hence forgotten, during the extinction phase[23,24]. However, experiments[25] and clinical observations[1] have challenged this view. Most importantly, seemingly extinguished memories can return unexpectedly, either after a sufficient period of time has elapsed (spontaneous recovery)[26], an explicit change of context (renewal)[25] or a presentation of the US on its own (reinstatement)[27]. In line with state-dependent accounts, these observations suggest that rather than forgetting or overwriting aversive associations, extinction involves the creation of a new memory together with the inhibition of the previous association[21]. One intriguing possibility which follows from this idea is that the individual may have learned that there are now two states that occur repeatedly (acquisition/high-threat state and extinction/low-threat state in the example). The same stimulus can therefore have different associations with outcomes depending on the state, explaining why animals and humans can suddenly behave differently in response to the same observation in the case of spontaneous recovery, reinstatement or renewal[28]. The notion of state-dependent learning is also in line with data showing that gradual, rather than abrupt, contingency changes lead to a decrease in the rates of spontaneous recovery, reinstatement and fear re-acquisition on a subsequent session[29–31], a phenomenon known as gradual extinction or occasional reinforced extinction. One study using continuous measure of state inference found that individuals who tend to represent the environment as multiple states have significantly higher rates of spontaneous recovery, as indexed by skin-conductance[32]. This suggests a relationship between state inference and spontaneous recovery of aversive associations.

While much research supports the general existence of a state inference mechanism, the question of which factors influence the creation of internal states, and how trait anxiety might relate to it, has remained less clear. First, the role of trait anxiety (TA) in state inference has not been explicitly tested, although some studies suggest such a link[31] (see also[33] for the same proposition in PTSD). High TA has been associated with an increased return of fear following phobia treatment[11,12] as well as with heightened neural and physiological differentiation between cues associated with a shock (CS+) vs no shock (CS-)[14,34,35]. Linking these findings to the theoretical work on state inference, it is conceivable that high TA individuals tend to learn in a more context-specific manner. If this context-specific learning is associated with a propensity to reactivate a previously experienced high-threat state, this could lead to repeated relapses as observed in patients.

Here, we investigate the context-specificity of learning in aversive environments and its modulation by trait anxiety. Our main hypothesis was that trait anxiety is associated with a higher propensity to associate periods of relative safety and harm with distinct internal contexts, and we argue that this process might explain persistence and recurrence of unwanted experiences, akin to relapse phenomena observed in clinical populations.

We employ a probabilistic aversive learning paradigm where the probability to receive a shock following a cue reversed regularly. In such an environment, participants can either update their expectations about the shock probability from trial-to-trial, or they can infer that there were in fact two contexts that correspond to relative safety and threat. We reason that participants with higher levels of trait anxiety are more prone to infer two distinct contexts rather than learn gradually. We test our hypothesis by comparing the performance of two computational models that capture the differences between gradual learning and context-dependent learning, and yield detailed predictions about trial-to-trial changes in behavior. Our results indicate that individuals with higher levels of trait anxiety tend to identify latent changes in the environment, as evidenced by steeper learning following environmental changes relative to oddballs, as well as model comparisons.

## Results

Our main objective was to study how learning about changing aversive outcome associations is affected by variations in trait anxiety among healthy participants. Eighty-nine participants (47 female, mean age: 25.5 years) performed a probabilistic aversive reversal learning task during which they saw one of three possible cues and were then asked to rate the probability of receiving a shock (Fig. 1a). The dataset was acquired in three separate experiments. Experiments I and II consisted of one session (75/25, see below), Experiment III was comprised of three sessions, with each session differing in outcome uncertainty. Therefore, the number of participants included in each session differed ($N_{60/40} = 36$; $N_{75/25} = 88$; $N_{90/10} = 37$; see Methods and Supplementary Materials for a detailed breakdown). In all three datasets, three different visual cues were each associated with a probability of receiving an electric shock. Two cues were consistently associated with either a high or a low shock probability throughout the session. We refer to these as stable-high and stable-low cues. The third cue started with a shock probability corresponding to either the stable-high or stable-low cue but reversed its probability a total of 6–10 times during each session (henceforth: reversal cue; reversals occurred randomly every 15.4 trials on average, see Fig. 1b, as well as Methods for details).

The three sessions varied in the amount of outcome uncertainty. In the 90/10 session, the stable-high probability cue was followed by a shock on 90% of trials, while the shock appeared on only 10% of trials after the stable-low probability cue. The reversal (i.e., changing) cue switched between phases of high (90%) and low (10%) shock probability. The 75/25 and 60/40 sessions followed the same logic: the reversal cue probabilities switched between 75% and 25% and between 60% and 40%, respectively. We use the term phase when referring to periods of fixed shock probability within the reversal cue. Participants in Experiment 3 completed all three sessions, while participants in Experiments 1 and 2 only completed a 75/25 session. The session order in Exp. 3 was counterbalanced across participants (Fig. 1c).

The shock intensity was individually calibrated at the beginning of each session to induce moderately high pain (rating of 8 on a numeric rating scale from 1–10, defined as painful but bearable considering the number of trials). The calibrated stimulus intensity did not differ between studies. There was no significant relationship between shock intensity and probability ratings, or between pain intensity and trait anxiety ($p > 0.05$, see Methods). Session order and initial shock probability of the reversal cue (high or low) were also found to have no significant effect (see Methods). Experiments did not differ with respect to mean stimulus intensity applied or participants' trait anxiety level. However, because several details of the experimental protocol differed between experiments, we decided to include Experiment as a factor (i.e., random effect) in all analyses. All participants completed the STAI-TRAIT questionnaire[36], which was used to assess the individual trait anxiety (TA) scores (median score: 39; range 20–71). While TA was included as a continuous parametric variable in all relevant

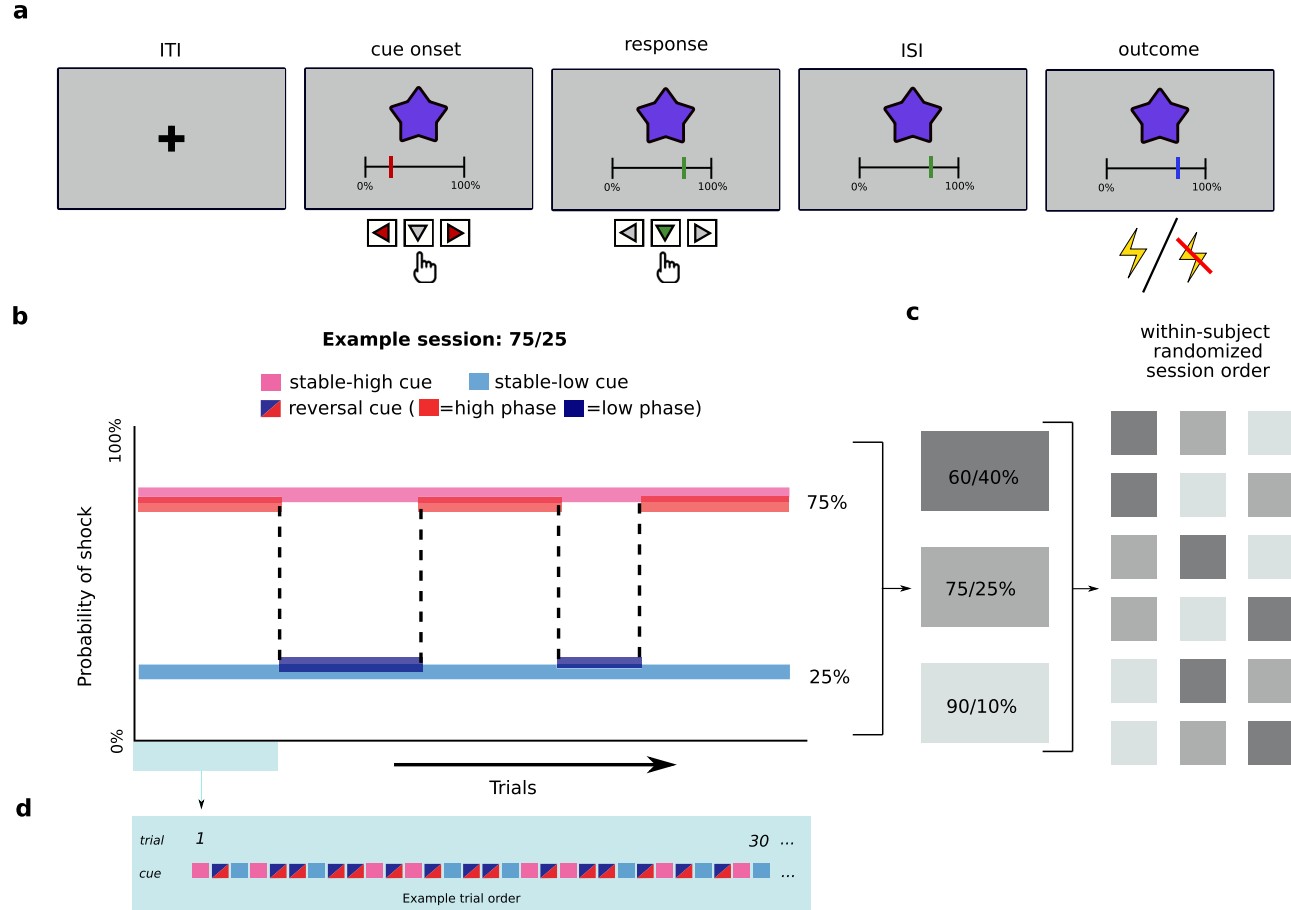

**Fig. 1 | Structure of the task and individual trials. a** Trial structure. Each trial started with a fixation cross (inter-trial interval; ITI) followed by one of the three cues (abstract fractals). Participants were asked to indicate the expected probability of receiving a shock on this trial by moving the red slider between 0% and 100% in increments of 1%. The final answer was submitted by pressing the downward arrow, after which the slider turned green to confirm the submission. After an inter-stimulus interval (ISI), a painful electrical stimulus (intensity: 8/10) was either delivered or omitted and the slider changed color to blue to indicate that the outcome had occurred. **b** Experimental design, example of one of the three sessions. The task was a continuous stream of trials. On each trial, one of the three cues was presented. While the shock contingency of the stable cues remained unchanged throughout the task (pink and light blue), the reversal cue changed every on average 15.4 trials between a low (dark blue) and a high (red) shock probability (i.e., phase). **c** In Experiment III participants completed the three sessions in a randomized order, in Studies I and II, only the 75/25 session was completed. **d** The three cues (stable-low, stable-high and reversal) were presented in a pseudo-randomized order. The reversal cue was presented on a larger proportion of trials than the stable cues. For details see Methods. Finger icons created by Rahul Kaklotar – Flaticon.

analyses, we sometimes report and visualize mean ratings per anxiety group (median split into high vs low trait anxious) in plots for illustration purposes.

Our first analysis focused on participants' ability to track shock contingencies associated with the stable cues. A linear mixed effects model (LMM) with probability ratings as a dependent measure revealed a main effect of cue type (stable-high vs stable-low cue), $F_{(1,308)} = 435.8$, $p < 0.001$, $\eta^2_p = 0.59$ [0.52, 0.64], with higher probability ratings for stable-high than stable-low cues (Fig. 2a). Although participants' ratings were relatively close to the true contingency levels, we found that participants slightly overestimated the shock probability for the stable-low cue (25% true vs. 30% estimated) and slightly underestimated the probability for the stable-high cue (75% true versus 71.8% estimated probability). Participants' probability ratings also reflected the contingency differences between sessions, as shown by the significant interaction between cue type and session type (90/10, 75/25 and 60/40), $F_{(1,308)} = 33.83$, $p < 0.001$, $\eta^2_p = 0.02$ [0.00, 0.06], see Fig. 2b (also Supp. Mat. for details).

Next, we asked whether participants' shock probability ratings for the two stable cues differed depending on trait anxiety. We averaged shock probability ratings per cue and participant, and ran a LMM with trait anxiety and session as fixed effects. This analysis revealed that the difference in ratings between high- and low-prob cues increased as a function of trait anxiety, as indicated by an interaction of TA and cue type, $F_{(1,308)} = 6.91$, $p = 0.009$, $\eta^2_p = 0.02$ [0.00, 0.06] (Fig. 2c). There was a positive association with TA in stable-high cue, $\beta = 0.0024$, and a negative relationship in the stable-low cue, $\beta = -0.0024$: the higher participants TA score, the higher reported ratings were in the stable-high condition, and the lower in the stable-low condition. Direct contrast of the associations of TA and rating between high and low-prob cues showed significant difference, $t_{(242)} = 2.63$, $p = 0.009$, $\eta^2_p = 0.03$ [0.00, 0.08]. We also tested whether ratings differed significantly from the true contingency level using one-way $t$-tests (Fig. 2d). When judging the stable-high cue, less anxious participants significantly underpredicted the true reinforcement level in the 75/25, $t_{(47)} = -2.62$, $p = 0.047$, $\eta^2_p = 0.13$ [0.01, 31], and 90/10 sessions, $t_{(18)} = -3.51$, $p = 0.015$, $\eta^2_p = 0.41$ [0.07, 65]. When judging the stable-low cue, less anxious participants overpredicted the probability in the 75/25 session, $t_{(47)} = 3.58$ $p = 0.010$, $\eta^2_p = 0.21$ [0.04, 40]. More anxious participants, in contrast, did not show over- or underpredictions, all $p$s > 0.05.

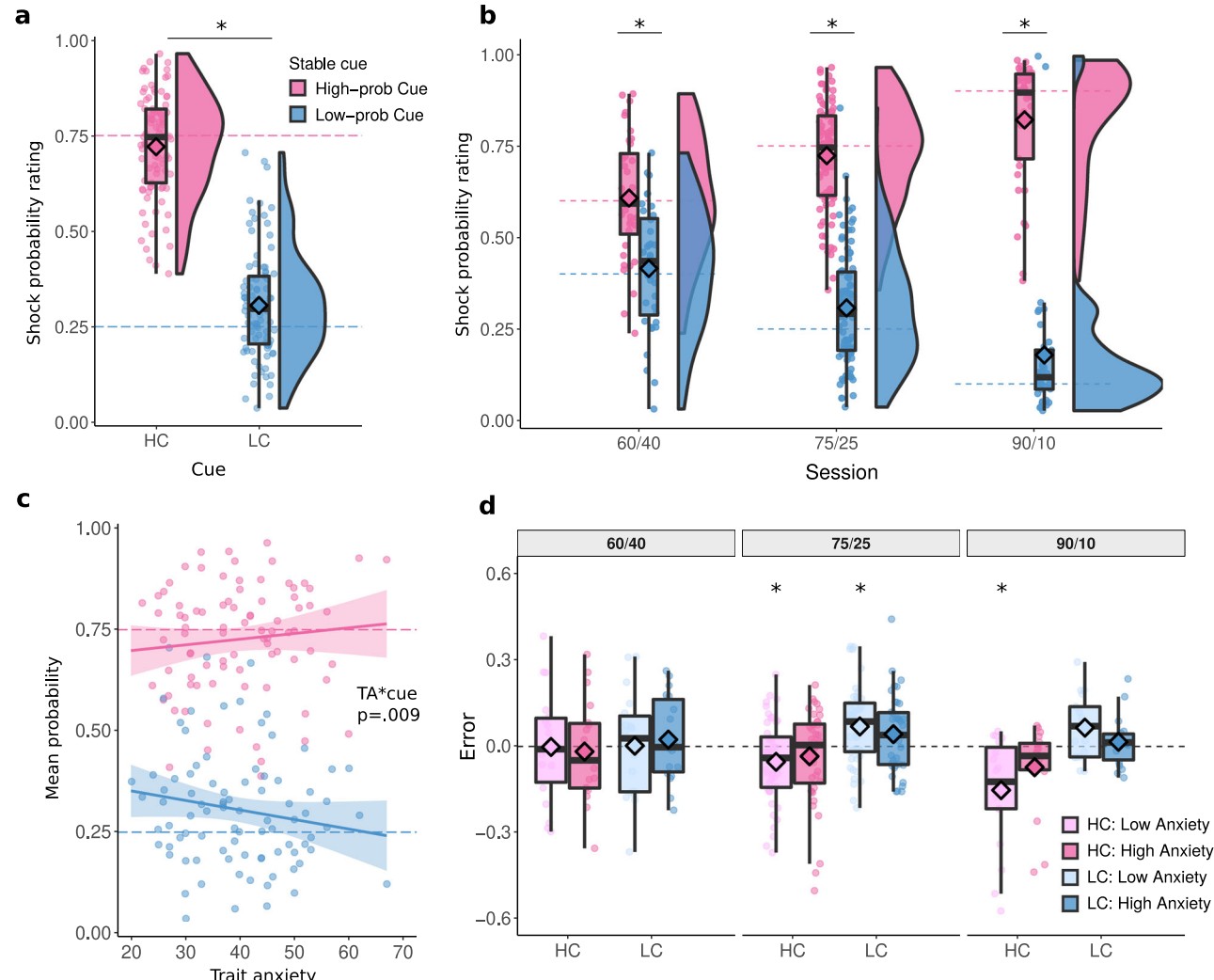

**Fig. 2 | Probability ratings for the two stable cues across all sessions.** Probability ratings were higher for the high-prob cue (pink) than the low-prob cue (light blue), F(1,308) = 435.8, $p < 0.001$, $\eta^2_p = 0.59$ [0.52, 0.64] across sessions, **a** and **b** in each of the three sessions, all $t$s > 4.9, $p$s < 0.001. **c** The difference between high and low stable cue increased with trait anxiety, t(242) = 2.63, $p = 0.009$, $\eta^2_p = 0.03$ [0.00, 08] All effects were assessed using LMMs, tests two-sided, post-hoc p-values Tukey corrected. **d** The divergence of ratings from the true reinforcement schedules (calculated as a cumulative mean) split by median TA, cue, and session. A positive error indicates an overestimation of shock probability, a negative error shows an underestimation. Low anxiety underpredicted true levels of stable-high cue in

75/25, t(47) = −2.62, p = 0.047, $\eta^2_p = 0.13$ [0.01, 31] and 90/10, t(18) = −3.51, p = 0.015, $\eta^2_p = 0.41$ [0.07, 65], and overpredicted stable-low cue in 75/25, t(47) = 3.57, p = 0.010, $\eta^2_p = 0.21$ [0.04, 40]. Asterisks indicate a significant difference from the true reinforcement level using one-way $t$-test, two-sided, p-value FDR-corrected. All panels include data for N = 89 participants, in panels **b** and **d** the three conditions include N = 36, N = 88 and N = 37 individuals. Box covers interquartile range (IQR), mid-line reflects median, whiskers the +/−1.5 IQR range. Horizontal dashed lines on all panels represent the true shock probability levels. Angled rectangles represent predictions of the fitted LMM model.

Next, we analyzed shock probability ratings for the reversal cue. This cue switched between phases of high and low-probability of shock (high-prob phase and low-prob phase). After each reversal, ratings tended to change rapidly, settling within about 5–10 trials at a stable level thereafter (see Fig. 3a). We first focused on participants' probability estimates during the stable periods between trial 10 and the next reversal (orange box in Fig. 3a), that is, after the initial learning period. Ratings provided during the high-prob phase were higher than during the low-prob phase across all three sessions (main effect of phase: F(1,234) = 165.92, $p < 0.001$; 63.0% vs 48.3% in 60/40, 71.4% vs 45.5% in 75/25 and 76.3% vs 38.4% in 90/10, high- vs low-prob phases respectively, Fig. 3a). Moreover, the model revealed an interaction between phase and session, F(2,234) = 8.86, $p < 0.001$, $\eta^2_p = 0.07$ [0.02, 0.14]. Post hoc tests found that this was driven by increased ratings of the high-prob phase in the 90/10 relative to the 60/40 session, t(229) = −3.40, $p = 0.010$, $\eta^2_p = 0.05$ [0.01, 0.11]. No credible evidence

was found for a difference between sessions in the low phase (See Supp. Mat.).

Our analysis also revealed a significant interaction between phase and TA, F(1,234) = 13.75, $p < 0.001$, $\eta^2_p = 0.06$ [0.01, 0.12] (Fig. 3b). Post-hoc tests found that this was driven by a significant negative relationship between TA and ratings in the low-prob phase, F(154) = 9.43, $p = 0.003$, $\eta^2_p = 0.08$ [0.00, 0.18], β = −0.0052 [−0.0085, −0018]. More specifically, low trait anxious participants overestimated the shock probability in the low probability phase by 23.7%, compared to 13.8% in high TA. No credible evidence was found for a difference in ratings in the high-prob phase, where low TA participants underpredicted the shock probability by 5.0% and high TA participants by 4.3% (TA was median split, see Fig. 3c for continuous relationship). This pattern was similar in all three sessions, although the association with TA was numerically most pronounced in the 90/10 session (interaction of TA with session: $p > 0.05$; Fig. 3c, Supp. Table 3).

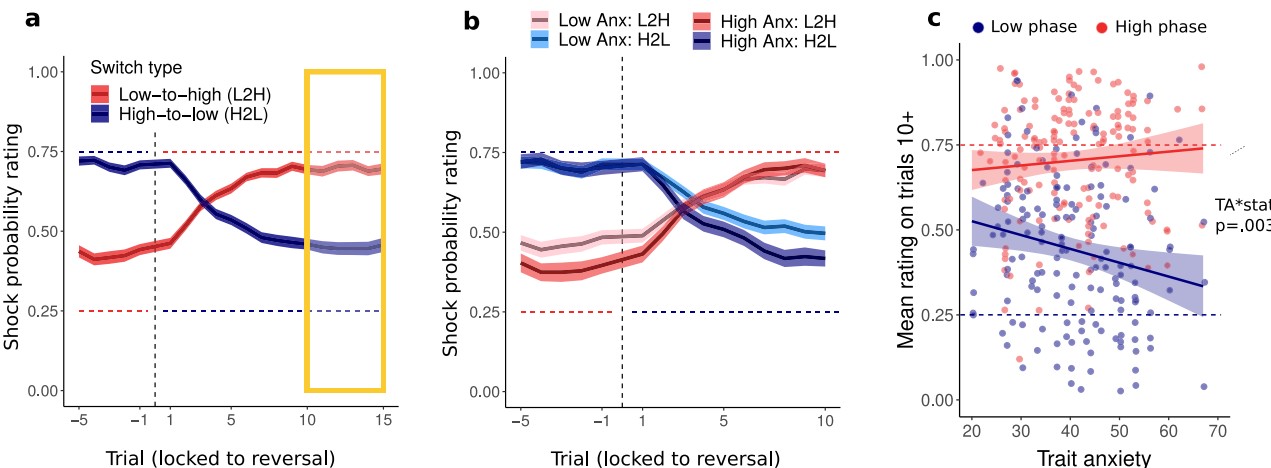

**Fig. 3 | Probability ratings in the reversal cue. a** Probability ratings locked to the reversal point for high-to-low (blue) and low-to-high (red) switches across the entire sample and **b** separately for high and low (median split) trait anxious participants. Trial indices indicate trials prior and following a reversal. **c** Probability ratings for trials 10 after current reversal until next reversal shown separately for each session as a function of trait anxiety. Ratings were higher in the high compared to low phase, F(1,234) = 165.9, $p$ < 0.001, $\eta^2_p$ = 0.41 [0.32, 40]. In the low phase, ratings negatively associated with TA, F(154) = 9.43, $p$ = 0.003, $\eta^2_p$ = 0.08 [0.00,

0.18], β = −0.0052 [−0.0085, −0018], The 10+ window was selected as the point from which shock ratings stabilized after the reversal. All assessed effects were by LMM, tests are two-sided, post-hoc p-values Tukey corrected. Horizontal dashed lines on all panels represent the true shock probabilities. Each panel contains data for all participants ($N$ = 89). Panels **a** and **b** show mean and standard error of the mean (shaded). Straight lines and shaded areas in panel **c** show per-condition linear fit y - x.

We next focused on the learning immediately after a reversal, i.e., trials 1 to 10 (reversal period). We characterized the speed of learning following a reversal by fitting a line to ratings on trials 1 to 10. This was done using a LMM with slope for each participant and phase. As expected, slopes differed depending on the direction of the switch, i.e., low-to-high switches were positive (2.44%; read as: the shock probability rating increased by 2.44% per trial) while slopes in high-to-low switches were negative (−2.33%).

Next, we took the absolute value of the slope estimates and included the values in a LMM model testing for effect of session, phase and TA on slopes. The statistical model found a main effect of session, F(2, 96) = 53.87, $p$ < 0.001, $\eta^2_p$ = 0.53 [0.39, 0.63]. Post-hoc comparisons between sessions revealed that mean steepness was significantly higher for 90/10 compared to 60/40 cues, $t_{60<90}$ = −9.84, $p$ < 0.001, $\eta^2_p$ = 0.15 [0.10, 0.21], and 75/25, $t_{75<90}$ = −7.19, $p$ < 0.001, $\eta^2_p$ = 0.38 [0.22, 0.51]. Furthermore, we found a positive main effect of TA, F(1,87) = 5.85, $p$ = 0.018, $\eta^2_p$ = 0.06 [0.00, 0.18] and a significant interaction between TA and session, F(2, 600) = 5.33, $p$ = 0.005, $\eta^2_p$ = 0.02 [0.00, 0.04]. The associations between TA and slope were positive in all sessions $\beta_{60/40}$ = 1.85 [−2.24, 5.93] × $10^{-4}$ $\beta_{75/25}$ = 1.10 [−1.98, 4.17] × $10^{-4}$ $\beta_{90/10}$ = 7.48 [3.45, 11.41] × $10^{-4}$. Post-hoc analyses found that this was driven by a significantly stronger association between TA and slope in the 90/10 session compared to 60/40, $t_{60<90}$ = −2.54, $p$ = 0.030, $\eta^2_p$ = 0.01 [0.00, 0.04], and 75/25, $t_{75<90}$ = −3.05, $p$ = 0.007, $\eta^2_p$ = 0.01 [0.00, 0.04]. Despite the overall main effect, the relationships between TA and slope in the 60/40, F(240) = 0.79, $p$ = 0.375, $\eta^2_p$ = 0.00 [0.00, 0.03], and 75/25, F(141) = 0.50, $p$ = 0.482, $\eta^2_p$ = 0.00 [0.00, 0.05], sessions were not significant, see Fig. 4. These results remained unchanged while controlling for the pre-reversal baseline (see Supp. Mat.).

Next, we reasoned that those participants who employed a state inference strategy should be better at knowing when to learn from outcomes, i.e., they should update less following oddball events and more just after contingencies have reversed. To test this, we calculated learning rates separately for five trials immediately after reversal (i.e. meaningful learning) and trials during the relatively stable periods between trial five and the next reversal (oddball trials, see Methods). Learning rates were log-transformed for the analysis (similarly to ref. 9). Learning was significantly higher after reversals(α = 0.265) compared to oddball trials(α = 0.225), F(1,534) = 12.85, $p$ < 0.001, $\eta^2_p$ = 0.03 [0.01, 0.07]. This effect interacted with TA, F(1,535) = 4.56, $p$ = 0.033 $\eta^2_p$ = 0.02 [0.00, 0.04]: TA was associated positively with learning on meaningful trials, β = .0015, CI95 = [−.0006.0037] and negatively, β = − .001, CI95 = [−.003 .002] with learning on oddball trials, statistical contrast of the two trends found a significant effect, t(528) = 2.136, $p$ = 0.033, $\eta^2_p$ = 0.02 [0.00, 0.04] (see Fig. 5a). Taken together, the behavioral analyses provide a link between trait anxiety and two behavioral markers of state inference: increase in slope following reversal, and reduced learning from oddball events during stable periods.

The results above suggested that trait anxiety is linked to faster updating of expectations when contingencies change. Such behavior could either be based on faster gradual learning or reflect state switching. To distinguish between these two ideas more formally, we fitted models to participants' probability ratings in of the reversal cue that assumed either gradual updating of a single state (1-state model) or updating of, and switching between, multiple states (n-state model). In both models, a state was characterized by a beta distribution that reflected the learner's current belief about shock probability. The 1-state model formalized gradual updating of a single beta distribution[37–39]. The shape of the distribution was controlled by parameters α and β, which were adjusted following each outcome. Specifically, following a shock α was updated using a step size parameter $\tau^+$, whereas β was increased by $\tau^-$ when no-shock was received. To model forgetting, in all trials both parameters also decayed by a free parameter λ, so that $\beta_{t+1} = \lambda\beta_t$ and $\alpha_{t+1} = \lambda\alpha_t$. The 1-state model captured gradual learning in a manner resembling the Rescorla-Wagner model, although it should be noted that the beta updating rule also incorporated accelerated learning rates following reversals, akin to Pearce Hall model (see Supp. Mat. for explicit comparison between 1-state learner and RW and PH). The n-state model started each session with a single beta distribution that was updated over time in a manner identically to the 1-state model. Crucially, however, the model kept track of recency-weighted surprise. If the surprise exceeded a threshold (controlled by a free parameter η), the model created a new state that minimized the current surprise. If more than one state already existed, the model first polled for existing states, and switched to them if a suitable candidate was found, before creating a new state. Every

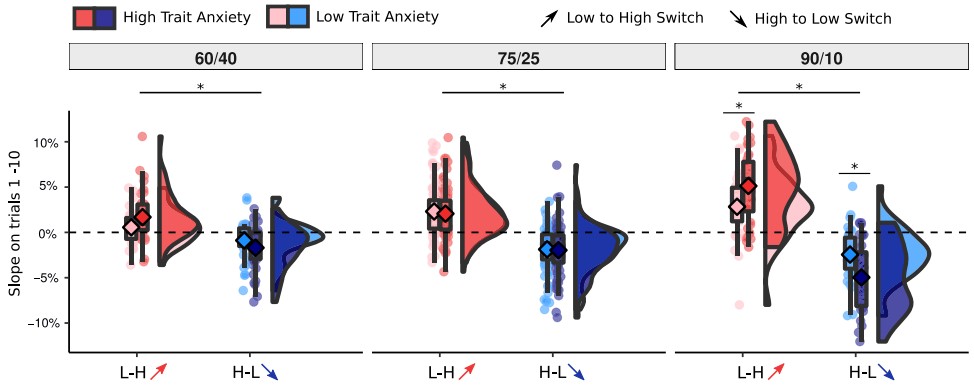

**Fig. 4 | Estimated slopes.** Slope of change of reported ratings on trials 1–10 following contingency reversal separately for each session and anxiety level. Anxiety was split by median for visualization purposes, but the statistical tests and corresponding asterisks reflect continuous relationships. Positive values indicate an increase in shock probability ratings while negative values indicate a decrease. The slope variable is shown using the original values, however, statistical tests were performed on absolute values. Slope was steeper in 90/10 compared to 60/40, $t_{60<90} = -9.84$, $p < 0.001$, $\eta^2_p = 0.15$ [0.10, 0.21], and 75/25, $t_{75<90} = -7.19$, $p < 0.001$,

$\eta^2_p = 0.38$ [0.22, 0.51]. Trait anxiety was positively associated with slope in the 90/10 condition, $F(1,233) = 13.39$, $p < 0.001$, $\eta^2_p = 0.05$ [0.01, 12] (across L-H and H-L), Assessed by LMM, tests two-sided, post-hoc p-values Tukey corrected. Asterisks indicate significant post-hoc tests. The three conditions include $N = 36$, $N = 88$ and $N = 37$ individuals. Box covers interquartile range (IQR), mid-line reflects median, whiskers the +/−1.5 IQR range. Angled rectangles represent predictions of the fitted LMM model.

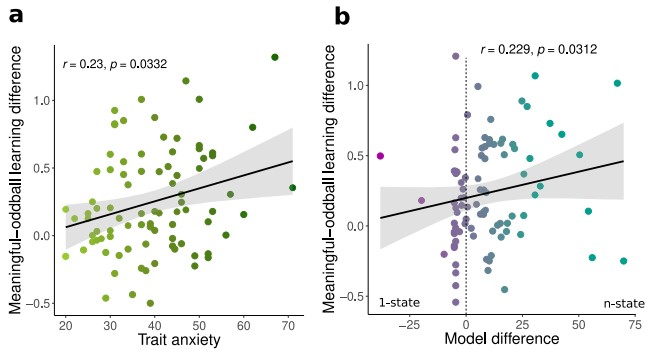

**Fig. 5 | Learning difference for meaningful and oddball events in trait anxiety and relative model fit. a** Trait anxiety was positively associated with relative learning from oddballs, $r(87) = 0.23$, $p = 0.033$, CI = [0.04, 0.41] across all sessions ($N = 89$). **b** Relative model fit was positively associated with relative learning from oddballs, $r(87) = 0.23$, CI = [0.01, 0.43], $p = 0.031$ across all sessions ($N = 89$). Both tests were performed using Spearman's correlation with bootstrapped confidence intervals. Statistical tests were performed on log-transformed learning rates.

time a new state was created the threshold for creating a new state increased. This allowed the model to generate a meaningful number of states and effectively switch between them. Both models are described in detail in Methods, and parameter estimates are summarized in Supplementary Tables 4a and 4b.

To assess which participants tended to learn gradually versus infer states, we calculated BIC scores for both models and subtracted them. Note that although the n-state model has one additional parameter (the threshold), it can behave almost identical to the 1-state model when the threshold is so large that the model never creates more than one state. Lower BIC scores of the n-state model can therefore be attributed to the necessity of inferring and switching states, i.e., that participants with improved model fit for the n-state over 1-state model are likely to rely on a state inference mechanism. Testing the model across all sessions, the n-state model fitted the data better (1-state BIC: −118; n-state BIC: −123). This was also true when comparing model fit for all three sessions individually, 60/40 (−84 vs −91), 75/25 (−133 vs −143) and 90/10 (−116 vs −132). However, the most pronounced difference was found in the 90/10 session where the n-state model improved fit substantially (see Fig. 6b). The same pattern emerged

when assessing the percentages of participants best fitted by each model: 41.7% vs 58.3% in 60/40; 43.2% vs. 56.8% in 75/25 and 37.8% vs 62.2% in 90/10 (1-state vs. n-state respectively). This suggests increased reliance on state inference in environments with larger switches in probabilities.

There was also considerable within-participant consistency of winning model across sessions. Namely, 38% participants (chance level: 11%) were fitted by the same model in all three sessions. Further 32% of participants relied on gradual leaning in 60/40 but switched to state inference strategy in 90/10. For a full breakdown of within-participant strategy by session see Supp. Mat.

Improved model fit should also be reflected in behavioral signatures of state switching. In line with this assumption, data of participants better fit by the 1-state than the n-state model exhibited more gradual learning, while steeper post-reversal learning was found in participants with better relative fits of the n-state model (Fig. 6a, see also Supp. Fig. 11). To quantify this impression, we assessed two major markers of state inference: post-reversal slope and learning from oddball events. First, we correlated the differences in model fit against the fitted slopes from participants' shock ratings. This revealed a significant positive association across all three sessions, $r(87) = 0.36$, $p < 0.001$, indicating that improved fit of the n-state model related to the steepness of estimated switches. Second, we analyzed the relative model fit in relation to learning from oddball events. Participants who were fitted better by the n-state learned more from outcomes occurring after reversal compared to oddballs ($\alpha_{meaningful-oddball} = 0.059$) while participants fitted better by the 1-state model had a smaller difference in learning rates ($\alpha_{meaningful-oddball} = 0.021$), $t(69) = -2.11$, $p = 0.039$, $\eta^2_p = 0.06$ [0.00, 0.20]. In a continuous analysis, the relative model fir correlated with reduced learning from oddballs compared to meaningful trials, $r(87) = 0.23$, CI = [0.01, 0.43], $p = 0.031$. Additionally, both behavioral markers of state inference (slope and meaningful-oddball learning rates) were tested using out-of-sample fits (fits from first half were related to behavioral data from second half). In both cases, the relationships remained significant. See Supp. Mat for details.

We next examined the relationship between trait anxiety (TA) and state inference by constructing a LMM with model fit difference as the dependent variable and TA and session as fixed effects. This model identified a significant interaction between TA and session, $F(2,105) = 5.20$, $p = 0.007$, $\eta^2_p = 0.09$ [0.01, 0.20]. Post-hoc analyses revealed that this was driven by a positive association between TA and fit improvement in the 90/10 session, $F(1,153) = 9.61$, $p = 0.002$

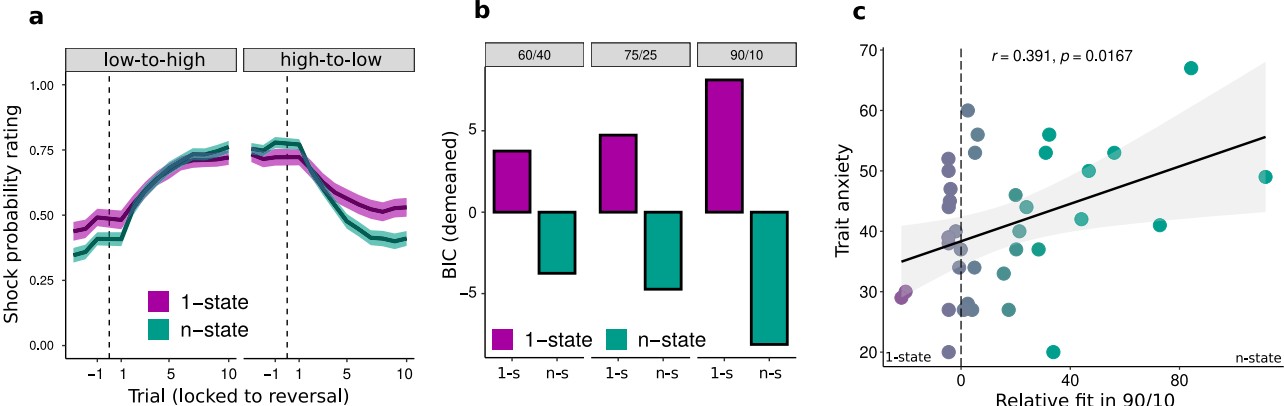

**Fig. 6 | Modelling results. a** Mean shock probability ratings separately for $N = 89$ participants better fitted by 1-state (purple) and n-state (green) models. The thick line denotes mean of each group, the shading reflects the standard error of the mean, **b** Mean BIC scores for the two models. BIC scores were demeaned to make sessions visually comparable. The conditions contain $N = 36$, $N = 88$ and $N = 37$ data points respectively. **c** Positive association between relative model fit and trait anxiety in the 90/10 session, $r(36) = 0.39$, CI = [0.07, 0.63], $p = 0.017$ ($p = 0.023$ for cross-validated correlation). The plot shows the result for $N = 37$ participants analyzed using Pearson's correlation, two-tailed, p-value Bonferroni-corrected for multiple comparisons.

$\eta^2_p = 0.06$ [0.01, 0.14] (see Fig. 6c). There was no significant association in the 60/40 or 75/25 sessions. Betas for the TA effect by condition were 0.04, −0.07 and 0.92 (60/40, 75/25 and 90/10). The association was significantly stronger in the 90/10 session compared to both 60/40, $t(78) = −2.64$, $p = 0.027$ $\eta^2_p = 0.08$ [0.00, 0.21] and 75/25, $t(111) = −2.96$, $p = 0.011$ $\eta^2_p = 0.07$ [0.01, 0.18]. This result was confirmed by a permutation-tested correlation between TA and model fit improvement which was significant in the 90/10 session, $r(36) = 0.39$, $p = 0.023$ (two-tailed, corrected for multiple comparisons). These results suggest that individuals high in trait anxiety tend to rely more on state inference.

To better understand the fitted n-state model, we explored its behavior and fitted parameters in more detail. We analyzed the internal number of states, model-estimated switchpoint, estimated uncertainty, step size parameters ($\tau^+$ / $\tau^-$), the threshold parameter ($\eta$), the decay parameter ($\lambda$) and the starting values of $\alpha$ and $\beta$. These analyses found a significant effect only in the fitted step sizes for the positive and negative outcomes $\tau^+$ and $\tau^-$. A LMM with parameter type ($\tau^+$ / $\tau^-$), TA and session as fixed effects found a significant main effect of parameter type, $F(1,228) = 37.03$, $p < 0.001$, $\eta^2_p = 0.14$ [0.07, 0.22], which reflected that shocks elicited larger updates than no-shocks $\tau^+ = 1.13$ vs. $\tau^- = 0.73$. There was no main effect of TA, or interaction of outcome type (shock/no-shock) and trait anxiety. Note that the same two parameters of the 1-state model, had a similar difference $\tau^+ = 1.17$ vs. $\tau^- = 0.86$), suggesting that differential learning from shock and no-shock events alone was unable to explain our behavioral effects of TA (see Supp. Mat. for full analyses of model parameters).

## Discussion

We investigated how outcome uncertainty and trait anxiety influence the tendency to infer and use hidden contexts in aversive probabilistic environments. We collected trial-by-trial shock probability ratings while the true latent contingencies changed between phases of high and low probability of receiving a shock. Our results show that trait anxiety was associated with behavioral markers of state inference and improved fits of the n-state model. In particular, participants high in TA showed faster changes in probability ratings after a reversal and less learning from oddball events. Both of these markers related to TA as well as preferential fit of the n-state model, providing a link between behavior and model fits that both suggest tendency towards state inference in high TA. Hence, we present evidence for a link between high trait anxiety and the tendency to infer hidden states, and to switch between them. Both TA and state inference have been independently

linked to the fear relapse. Trait anxiety is known to be associated with higher rates of fear relapse[11,12] While representing environments as multiple states leads to higher rates of spontaneous recovery[32]. Our findings suggest that trait anxiety, as a time-invariant disposition[40], facilitates the parcellation of observations into different states that are characterized by different cue-outcome contingencies. In the clinical literature, the assumption of several independent states has been discussed in relation to prevention of updating of existing cue-outcome associations (i.e., overwriting previous memories) and thereby to hinderance of effective fear extinction[1]. Instead of revising the current situation, the individual assumes an additional new state that reflects the altered contingencies. Given the links between orbitofrontal cortex (OFC) the representation of task states[16,18,41] and anxiety[42], our finding also motivates future investigation into the role of anxiety in task-state representations in the OFC.

We note that the two learning mechanisms (state inference versus gradual learning) are not mutually exclusive, but might rather reflect different degrees of state-dependent learning (as in ref. 32). Our results indicate that the propensity for state-dependent learning might depend on the amount of outcome uncertainty in the environment, since better fits of the n-state model were observed in sessions with more distinct high- and low-probability states (90% and 10%), as compared to sessions with less distinct states.

The results corroborate previous theoretical predictions. In particular, parcellation into separate states (or contexts) was proposed to be associated with anxiety disorders and account for relapse phenomena[21,32,33]. Here, it is assumed that extinction involves learning that is separate and more complex from acquisition. The memory for acquisition needs to be inhibited during extinction while a new association needs to be learned[21]. In our behavioral results, the effects of anxiety were driven by more accurate probability estimates in the stable-low cue and the low-state of the reversal cue, which both correspond to conditions of relative safety. This aligns with some previous reports. For example, in a gamified aversive learning paradigm[37] reported a positive association between safety learning and state anxiety. Similarly, another study found that under a condition of higher cognitive load fear extinction (indexed by SCRs) was more successful in the high TA group[43]. Interestingly, high anxiety (i.e., factor loading high on trait anxiety) was associated with increased engagement of cognitive control in a go/no-go paradigm[44], suggesting that non-clinical TA might be associated with better use of cognitive resources. The broader literature on the relationship between trait anxiety and fear yields mixed results. While some studies report

increased discrimination of CS+ and CS-[35] and comparable fear inhibition during extinction in high vs. low TA individuals[45,46], others report deficits in inhibitory processing[6,47,48] and safety learning[14,49]. Taken together, subclinical levels of anxiety might be beneficial to aversive learning only under certain conditions, similarly to how it is sometimes beneficial in cognitive test performance[50]. This would be in line with the evolutionary conceptualization of anxiety as a state of increased vigilance to detect and avoid threats[51]. This diversity of findings has not been reconciled, however, a possible explanation in terms of methodological differences (e.g., modality, aversiveness, outcome uncertainty) has been suggested[46]. Additionally, previous work has identified lower TA in studies involving MRI[52]. Therefore, the sample of study 1 (fMRI study) might not be fully representative in terms of trait anxiety. However, the mean TA level was lowest in the drug study (Study 1: 40.3; Study 2: 37.3; Study 3: 41.4) which also showed the weakest behavioral effect as reported in Supp. Mat.

An above-chance proportion of participants (34%) switched from using gradual strategy in 60/40 to using state inference in 90/10. The optimal strategy for a given environment might depend on the tradeoff between cognitive effort and accuracy in prediction. Separating the environment into latent states arguably comes at higher cognitive cost which might only be justified in environments with low outcome uncertainty. Trying to infer states in noisy environments can require substantial cognitive load (e.g., integration over longer periods of time) and it might not yield much benefit (i.e., predictive accuracy). A major question that remains to be answered is why do high TA individuals rely on state inference mostly in the 90/10 session. One possibility might be that while the uncertainty in the noisy environments is too high and learning meaningful changes from stochastic events poses high cognitive demands, in 90/10, learning the structure of the environment can meaningfully result in reduction of internal uncertainty. An interesting future direction would be to investigate whether clinically anxious individuals continue to (perhaps sub-optimally) try and find structures in noisy environments, i.e., whether they tend to erroneously find too many latent causes, or whether they are instead driven by the adversity of high uncertainty itself and lump all experiences under a single latent cause (as in ref. 33).

How and whether states are inferred depends on uncertainty. In our data, state inference was most favored in environments where objective changes in shock contingencies were largest (90/10), i.e., when outcome uncertainty was low. However, in changing environments, inferring whether the objective probability has changed also depends on higher order uncertainty such as volatility[53] which has been previously associated with anxiety and depression[9,54]. When receiving a surprising outcome, one must consider both outcome uncertainty and volatility to determine whether it reflects change in state or an oddball event[10,55]. Interestingly, high TA has previously been associated with the inability to adjust learning to environmental volatility, as reflected in a high learning rate despite stable contingencies[9]. Our results show that high TA adjust their expectations faster. This might seem at odds with[9], however, it is important to consider methodological differences, such as using instrumental, rather than Pavlovian, learning to manipulate between-session, rather than within-session, volatility. Additionally, a recent study demonstrated how misestimation of outcome uncertainty (stochasticity) rather than volatility can drive learning and cause fast, jump-like learning from rare events due to misestimation of stochasticity[10]. To test whether such misestimation could explain our findings (as opposed to state inference), we compared learning rates during the period just after a reversal (meaningful learning) with learning rates in relatively stable periods, where unexpected outcome did not signal a reversal but rather an exception that was to be ignored (oddball learning), see Fig. 5. We argue that a bigger difference in learning rates for meaningful and oddball events reflects state awareness, that is, ignoring oddball events suggests knowledge of a higher-order structure. In our behavioral

results high TA individuals indeed showed decreased learning from oddballs. It should be noted, however, that a modulation of learning can in principle have many different sources[56–58], and differential treatment of oddball versus reversal events can also be accounted for in other implementations of state inference[59].

As mentioned above, our results diverge from previous findings that reported lack of fear inhibition during extinction[45] and deficits in safety learning[49] in high trait anxiety. A number of methodological differences compared to our study could account for these differences. Our study aimed to investigate a temporally extended learning process, which contrasts with the fear extinction paradigm used in the above-named studies. In particular, we used a task in which both acquisition (phase of high shock probability) and extinction (phase of low shock probability) are probabilistic. Notably, in the low phase, probabilities ranged between 10% and 40% whereas in other studies, this range was used during acquisition, i.e. the high-threat state[60,61]. Our choice was in part motivated by the importance of keeping the degree of outcome uncertainty identical in both phases within each session. As recently pointed out, uncertainty is a confound in studies where acquisition is probabilistic (e.g., 50% shock) but extinction is deterministic (0%)[62] (see also Discussion in ref. 46). Another important factor is that in our design, phases of high and low shock probability occur repeatedly as each participant experienced at least six contingency switches. This decision was again motivated by real-world conditions where aversive stimuli often reoccur (e.g., periods of back pain, exam stress). Our focus was to understand how individuals with varying degrees of trait anxiety intrinsically learn and represent the structure of an aversive environment which sets the study apart from classical studies on acquisition and extinction. However, future research should systematically investigate the role of trait anxiety under different relative conditions of threat in a manner similar to[63], including the difference in probabilistic versus deterministic environments.

One question that remains to be answered is whether the ratings-based results reported here would be followed by physiological measures. Physiological markers could mirror expectancy ratings and thereby reflect the individual's cognitive model of the environment[64,65] or they could reflect deliberate cognitive processes whereas the physiological fear response might diverge (as in refs. 45,66,67). For example, one study reported change in physiological responses following a reversal without contingency awareness[68]. In our case, high TA participants might be aware of relative safety (i.e., be in a subjective low state) but not be able to inhibit fear response. In support of this idea, a recent study looked at the relationship between SCR-indexed spontaneous recovery and state inference. In one of their analyses, they report that trait anxiety was not associated with SCR-indexed inference of multiple states[32].

A noteworthy aspect of our work is the model that captures state inference and updating. It combines single-state updating models under a beta distribution[37–39] with state inference models proposed previously[22,69,70]. The key feature of the model is that it can translate binary outcomes into probabilistic states, quantifying the current expectation and its uncertainty in the process (see Methods). We showed that the n-state model was able to estimate the appropriate number of states and that model-estimated switches occurred in the same period as in the behavioral data (see Supp. Mat.). Most importantly, there was a clear behavioral distinction between participants better fitted by the 1- versus n-state models (see Fig. 6a and Supp. Fig. 11).

Despite the model performing well for our purpose, it might require adjustments in other paradigms depending on the task and data. For example, in our version the mechanisms by which the model switches between states as opposed to creating new states are codependent and only differ in the difficulty parameter (i.e., more surprise is needed to create a new state with increasing number of states).

Future implementations of this model could include entirely separate thresholds for state switching versus state inference, for instance to study whether some groups tend to create too many states but never switch to a existing state.

Taken together, our results suggest that more trait anxious individuals have a tendency to represent aversive environments involving high and low threat contexts as distinct states and to incorporate this knowledge into their predictions. We suggest that this parcellation of the environment into states may explain previously observed fear relapse phenomena associated with trait anxiety and in turn contribute to treatment of anxiety disorders[71].

## Methods

The three studies included in this work comply with and were approved by the Central University Research Ethics Committee (CUREC) of Oxford University (R44738/RE001, R29583/RE004, R52892/RE001). All participants provided informed consent. We presented a pooled analysis of three studies that used a probabilistic aversive learning task (see *Task*). Experiment I is an fMRI experiment consisting of a short screening session and a main session 1–3 days later. Only data from the main session were included in this analysis. Experiment II is a three-visits (visit 1: baseline, visit 2: drug administration; visit 3: follow-up) drug study investigating the role of the angiotensin-II inhibitor drug losartan in aversive learning. Only the placebo group from the second visit was included in this analysis as the task on visits 1 and 3 was shorter. This ensures that participants in all three studies had the most similar experience. For a detailed overview of the three studies see Supplementary Table I.

Data were collected in the 3 Tesla MRI scanner of the Wellcome Centre for Integrative Neuroimaging (Experiment I), in behavioral testing laboratories of the Nuffield Department of Clinical Neurosciences (Experiment III) (John Radcliffe Hospital, Oxford) and the Department of Psychiatry, Warneford Hospital (Experiment II), Oxford. The factor experiment was included as a random effect in all main analyses.

### Power analysis

Following the identification of behavioral effect in Experiment I we designed Experiment III using the appropriate power calculations. Our main effect of interest was the correlation between trait anxiety and the difference in BIC scores for the two models. To detect a 0.5 effect size at 0.8 power at least within one of the three sessions (two tailed, Bonferroni corrected alpha = 0.0167) the sample of 38 participants is required. We therefore collected slightly more participants (40) in the study to meet that target.

### Participants

Eighty-nine participants (47 female, mean age: 25.5 years) were included in the data set (Experiment I: $N$ = 30, 16 female, mean age = 25.5; Experiment II: $N$ = 22, 10 female, mean age = 24.6; Experiment I: $N$ = 37, 21 female, mean age = 25.7). Gender of participants was based on self-report. It was not considered as a factor in the analyses because we did not have a specific hypothesis and because it was not related to trait anxiety levels. Participants for all three studies were recruited using local advertisement and the SONA recruitment system managed by the Department of Experimental Psychology, University of Oxford. Inclusion criteria varied slightly between studies (due to MRI data collection in Experiment I and drug administration in Experiment II). A comprehensive list of criteria can be found in the Supplementary Materials (section Inclusion and exclusion criteria). All studies included right-handed healthy adults aged between 18 and 40 years without a history of psychiatric illness and not taking any psychoactive medication (including recreational drugs) at least 3 months prior to the experimental session. In line with recent recommendations for exclusion criteria in aversive learning studies[72], data of all participants, including

non-learners, were included in the analyses. Participants were reimbursed 40 GBP in Experiment I (two visits, the second visit was a 2 h MRI scanning session), 60 GBP in Experiment II (three visits lasting 0.5, 3–3.5 and 0.5 h) and 25 GBP in Experiment III (single visit lasting 1.5 h).

In total, 116 participants took part in the three studies. Four participants were excluded due to missing behavioral data (presentation computer or shock administration stopped working properly), two because of missing anxiety scores (both participants had to leave the lab before completing the questionnaire) and one for misunderstanding the task. Twenty-two participants who received the drug in Experiment II were not included because questionnaire and physiological measures necessary to control for the effect of the drug were not available in the other two studies. In two participants, data of one of three sessions was missing due to script or stimulation failure. In this case, we included data of the remaining sessions in the analyses. Note, however, that therefore the degrees of freedom vary between sessions. The final number of participants included in the analyses was 89 (87 without any missing sessions).

### Aversive stimuli

Electrical stimuli were applied using a commercial electric stimulation device (Constant Current Stimulator, model DS7A; Digitimer, Hertfordshire, UK), delivering a 2 ms monopolar square waveform pulse via a concentric silver chloride electrode attached to the back of the left hand.

The stimuli were calibrated individually at the beginning of the task and during any pauses (Experiment I – every 13 to 18 min; Experiment II – every 12 to 15 min; Experiment III – just once at the beginning of each session–every 20 to 25 min). The target intensity was 8 on a scale ranging from 0 (not painful) to 10 (too painful to take part) scale. The 8/10 pain level was defined as a sensation that is painful but tolerable for a given number of trials (study-specific number corresponding to 50% of trials). Three qualitative anchor points were defined to help standardize the calibration across participants and studies: 1/10 which was defined as the intensity at which the sensation starts to be moderately painful (pain threshold); 8/10 is a sensation that is clearly painful but tolerable; and 10/10 which would be the level of pain which is too strong to be tolerated. The calibration followed the Method of Limits[73]. The stimulus intensity started at the pre-calibrated 1/10 level and changed after each rating in an increasing trend (individual stimuli could however get stronger or weaker). Upon each stimulus delivery, participants were asked to report how painful the sensation was on a rating scale ranging from 1 to 10. When a rating was higher than 8, the next stimulus was always lower. The calibration terminated once three out of the last five stimuli were rated as exactly 8. To ensure that the intensity remains at a subjective 8/10 level, regular re-calibrations took place.

### Task

The goal of the study was to investigate how participants learn to predict the probability of an aversive event and how they update their expectations on a trial-to-trial basis. To this end, we used a Pavlovian probabilistic learning task in which participants learned to associate three visual cues (abstract fractals, selected randomly for each participant from a pool of 20 possible fractals) with the delivery or omission of a painful electrical stimulus (shock). On each trial, participants were presented with one of the cues which could be followed by the electrical stimulation and asked to submit an expectancy rating. Throughout the experiment, one of the cues was followed by a shock on a high proportion of trials (60% to 90%) while no stimulus was applied in the remaining trials (stable high-prob cue). For the second cue, contingencies were reversed, i.e., the electrical stimulus was applied in a low proportion (10% to 40%) of trials (stable low-prob cue). For the third cue, shock contingency switched between the low and high probability in semi-regular intervals, mean 15.3 trials. Since our

primary analysis goal was to study how people learn about changes in contingencies, we designed the task in a way that the reversal cue appeared more often than the stable cues in all three studies (see Supp. Table I).

**Standard trial structure.** Each trial started with the presentation of a fixation cross (inter-trial-interval, ITI; Experiment I: 3–5 s; Experiment II: 2 s; Experiment III: 1–2 s). Next, the cue for this trial was presented and the participant had 4 s to submit a response. The fractal was shown in the middle of the screen while the slider used to provide the rating was positioned below. Using the left and right arrow keys (MRI button box in Experiment I), participants could move the slider on a scale from 0% to 100% (in increments of 1%). Once the desired position was reached, they could confirm and submit their rating by pressing the down arrow key (middle button on the MRI button box). Participants had up to 4 s to submit the rating. If rating was not submitted on time a warning message appeared and the trial was restarted. Once the rating had been submitted, the slider changed color to green. After an interstimulus interval (ISI; Experiment I: 2– 4 s; Experiment II: 1 s; Experiment III: 1–2 s) the outcome was delivered (i.e., shock delivery or omission). The outcome was accompanied by a change in the color of the slider to blue (to make timing of outcome equally clear to participants for both shock and no-shock trials) in Experiment III (in Experiments I and II the slider did not change color). The cue remained on the screen for additional 2 s (Experiment III: 1.5 s) and disappeared with the onset of the next ITI. See Fig. 1. Three fractals out of a pool of twenty were assigned randomly to the three cue conditions (stable-low, stable-high, reversal). The background color was gray (rgb = [0.71, 071, 0.71]) and this stimulus occupied 9 degrees of visual angle. The rating scale was shown just below the fractal. Only the two ends of the 0% to 100% expectancy rating scale were labeled by ticks. The slider was initiated at a random position on each trial.

**Bonus trials.** In Experiments I and II, participants were occasionally presented with two of the cues and asked to select the one with either lower or higher probability of shock. Unbeknown to the participants, there was always one cue with currently low (i.e., stable-low or reversal in low-prob phase) and one with high (i.e., stable-high or reversal in high-prob phase) probability. In Experiment III, on a similarly small proportion of trials, participants could wager an amount between £0 and £5 to avoid a single shock on the next trial. Both tasks were introduced to keep participants engaged and to obtain an additional measure of value. Due to the different nature of the ratings, analysis of this data was not included in the present work.

**Task structure.** The task was characterized by changes in the contingency of the reversal cue (switches) which occurred in irregular intervals (see Supp. Table I). We use the term phase to refer to a section of the task during which the shock probability of the reversal cue was constant. Each participant experienced between 5 to 9 switches which results in 6 to 10 phases per participant. Phases where the probability of the reversal cue was low are referred to as low-prob phase while phases with high probability are called high-prob phase. The number of trials and the dispersion of the switch point was slightly different in the three studies: Experiment I (M = 30, +/−2 trials), Experiment II (M = 30, +/−5 trials), Experiment III (M = 35, +/−10 trials). The mean refers to the total number of trials across all three cues. The proportion of stable-low, stable-high and reversal trials was as follows: Experiment I: 30%-30%-40%; Experiment II: 25%−25%−50%; Experiment III: 20%−20%−60%. This means that for example in Experiment III it was on average 0.6 × 35 = 21 reversal trials/phase, although with higher variability (minimum 15 trials). Individual trials were presented in pseudorandomized orders. The schedules were generated as follows. First, the number of trials for a given phase was determined (by phase we mean the period during which the reversal cue did not switch). Next, it

was ensured that the contingency of each of the cues was within +/−5% of the target. This means that if there were for example 40 trials in total, out of which 10 were stable-low, 10 were stable-high and 20 were the reversal cue, the target objective probabilities were 25%, 75% and 75% (assuming the 75/25 session and the reversal cue being in the high-phase) it was ensured that the objective shock rates delivered for each cue were within +/−5% of these contingencies, e.g., for the reversal cue there were between 14 (70%) and 16 (80%) shock trials within this mini block. Additionally, it was ensured that within each phase a given cue was not presented on more than three subsequent trials. For the reversal cue, on the first five trials after reversal, at least three outcomes were in the new direction (if switch from high-to-low phase just happened, at least three out of the first five trials ended with no-shock). Furthermore, once each mini-block passed the above criteria, the mini-blocks were assembled into a schedule. There was a slight difference between studies. While in studies I and II a change in phase occurred with 75% probability (i.e. sometimes it didn't happen), in Experiment III switch always happened. Lastly, a second +/−5% contingency check was performed, this time across the entire trial schedule separately for each cue.

## Instructions

To minimize any influence of the experimenter, the information about the task was presented in writing. Only if the participant required further explanation, instructions were clarified verbally according to protocolled answers. Participants were presented with minimal information regarding the number of cues, task duration, cue frequency and switches. They were told that each cue is associated with a certain probability of receiving a painful stimulus and to pay attention to all three cues as any of them may or may not change their probability signaling the painful stimulation at any point. Participants were also explicitly told that their ratings do not impact the outcomes. For details on the instructions see Supplementary Materials (Instructions section).

## Questionnaires

Trait anxiety was assessed using the STAI-TRAIT[36]. Additional study-specific personality measures were collected (e.g., pain-related fears and attitudes in Experiment I). For the complete list of questionnaires see Supplementary Materials, section Questionnaires. In Experiments I and III, questionnaires were completed using a computerized interface based on the LimeSurvey software. In Experiment II, pen & paper versions of the questionnaires were used.

## Data analyses

Data were analyzed using custom MATLAB[74] and R 3.6.3. scripts (for a complete list of packages and versions see associated repository). Stimuli were presented using MATLAB 2016a and Psychtoolbox 3[75]. Questionnaire responses were collected using LimeSurvey.

**Statistical and visualization approach.** Statistical analyses were performed using Linear Mixed Models (LMMs, as implemented in lmer 1.1–25 R package[76] with study and participant included as an effect with a random intercept. For each analysis we included fixed effect of interest also as a random slope, and we performed a model comparison between the two version. Adding random slope didn't result in improved model fit in any of the analyses, so we didn't include them. Following ANOVA analysis of LMM results, post-hoc tests are reported using corrected p-value (Tukey). Reported effect sizes use the partial eta squared metric with the corresponding 95% confidence intervals. Where variables were continuous (e.g., trait anxiety) they were included as such in the statistical models. All tests were two-sided at alpha = 0.05. To visualize data, we include raw data, summary statistics (mean or median), information about variance (standard error or interquartile range) and density (raincloud plots[77]. In time-series plots (e.g.,

Fig. 3a) we plot mean per condition and SEM (standard error of the mean). All analyses were tested for homogeneity of variance (KS-test) and visually inspected for normality of residuals (histogram). One model violated assumptions, the LMM testing for effect of oddball/meaningful trial type on model-free learning rates. We log-transformed the learning rates (the analysis that we actually report) which improved the homogeneity of variance but it didn't remove its violation entirely. We also replicated the result using beta regression. Both approaches resulted in the same set of results. Additionally, we note the robustness of LMM models to moderate violations in distributional assumptions, specifically heteroscedasticity[78].

**Computational modeling.** All models were fitted to the trial-by-trial shock expectancy data using Bayesian Adaptive Direct Search (BADS)[79] by minimizing the negative log likelihood of the data given a model. Our 1- and n-state models naturally use beta likelihood. To assess model fit across all trials, BIC[80] scores were calculated. To prevent convergence to local extremes, fitting was performed 45 times for each participant and cue, ensuring that computational resources were identical across all models.

## Measures
**Behavioral measures.** All main analyses including model fitting are based on shock probability ratings (0% to 100%) provided by the participant on each trial. Additionally, each study contained either cue preference ratings or a shock wagering task to provide additional measure of shock expectancy. However, because the measures varied across studies, they are not included. Lastly, at the end of the task we collected visual and general liking ratings for each of the images used in the task. Participants were presented with the three fractals and asked to rate their visual appeal and general liking' on a scale from 1 to 10.

**Slope after reversal.** To calculate the average speed of updating after reversal, we fitted the shock probability ratings data on trials 1 to 10 (period of change) using a linear mixed effect model with estimated slope for each participant, session, half (early/late) and switch type (high-to-low, low-to-high). Due to convergence issues, such model was fitted separately for each session, half and switch type. The estimates slopes for each participant/session were then extracted from the models and analyzed separately using another LMM.

**Error from true reinforcement.** To evaluate how much the individual learning time courses deviated from the delivered rates of shock, we calculated the running mean reinforcement rate (mean over shocks = 1 and noshock = 0 outcomes) for each state condition separately. This measure serves as an estimate of the true shock probability under the assumption that the agent knows which state they are in. To obtain a directional measure of error, the true reinforcement rate was subtracted from the expectancy ratings.

**Model-free learning rates in meaningful and oddball" trials.** To obtain trial-wise learning rates, we rearranged the Rescorla-Wagner (Eq. 1) learning rule and calculated the trial-specific learning rate $\alpha$ (Eq. 2), where $P$ stands for probability ratings, $O$ for outcomes (shock/no-shock) and $t$ for a given trial.

$$P_{t+1} = P_t + \alpha_t(O_{i,t} - P_t) \qquad (1)$$

$$\alpha_t = \frac{P_{t+1} - P_t}{O_t - P_t} \qquad (2)$$

where $0 \le \alpha_t \le 1$

In some cases, such calculated learning rates became negative, for example, when the participant received a shock, but they lowered their expectation. In this instance, ratings were excluded from the analysis (assigned NaN values).

To distinguish between learning immediately after reversal, when learning rates should be relatively higher (meaningful learning), and later in stable periods of each state, when learning from surprising events should be relatively slower (oddball learning), we split model free learning rates at fifth trial after reversal. For example, if shock occurred in the first five trials after low-to-high switch then it was considered meaningful to learn from, while if it occurred after fifth trial of high-to-low switch it was considered an oddball.

To check that our specific choice of post-reversal cutoff trial (ct = 5) did not drive the results, we calculated oddball/meaningful learning rates for three additional cutoff values: 7, 10 and 13. We next tested the impact of the cutoff threshold on the estimated meaningful/cutoff values using a LMM. We found no significant impact of the cutoff, all $p$s > 0.9. We also present the result in Supp. Fig. 3.

## Computational models
Our primary goal was to provide a set of two models which used the same updating mechanisms and distributional assumptions (beta distribution), and that differed only in the ability to infer states. To model gradual learning and switching, we used a framework based on the beta distribution, similarly to previous studies[37,38]. Our goal was to model the current shock probability estimate (ranging between 0 and 1) based on the received binary outcomes using the beta distribution. This approach is well-suited to model probabilities because beta distribution is bounded by 0 and 1. Additionally, it implicitly quantifies the amount of uncertainty about the current state. Lastly, this probability distribution naturally arises from binary outcomes. This provides a logical link between the outcomes delivered in the task (shock/no-shock) and the data reported by participants (probability estimates) but stands in contrast to more commonly used Normal distribution, which offers no straightforward mapping between binary outcomes and probability density.

**1-state model.** Each state was characterized by a beta distribution with parameters and $\alpha$ and $\beta$ (Eq. 3).

$$Beta_{PDF}(\alpha,\beta) = \frac{x^{\alpha-1}(1-x)^{\beta-1}}{\frac{\Gamma(\alpha)\Gamma(\beta)}{\Gamma(\alpha+\beta)}} \qquad (3)$$

Given this distribution, we assumed that the reported subjective probability of a shock reflected the mode (Eq. 4a) of the probability density function provided above, while state uncertainty was defined as standard deviation of the same distribution (Eq. 4b).

$$\hat{P} = \frac{\alpha - 1}{\alpha + \beta - 2} \qquad (4a)$$

$$\sigma = \sqrt{\frac{\alpha\beta}{(\alpha+\beta)^2(\alpha+\beta+1)}} \qquad (4b)$$

Parameters $\alpha$ and $\beta$ can be thought of as proportional to the number of shocks and no-shocks received up until this point. As the sum of $\alpha$ and $\beta$ increases, the variance (and therefore state uncertainty) of the distribution decreases. In other words, the more evidence is available to the model, the more certain it is about its probability estimate. The starting values of $\alpha$ and $\beta$ are estimated as free parameters ($\alpha_0, \beta_0$, both $\in [1,10]$, values smaller than 1 were not included because in this case the distributions become bimodal, and Eq. 4b). On each trial, the two parameters are updated by the amount equal to shock and no-shock attention weights $\tau^+$ or $\tau^-$ (both $\in [0,2]$) depending on whether the shock was received (+) or omitted (−). Specifically, if the cue was followed by a shock, then $\alpha$ is updated by

the amount of $\tau^+$ and if no-shock occurred then $\beta$ is updated by $\tau^-$. Additionally, on each trial both $\alpha$ and $\beta$ are subject to decay $\lambda \in (0,1)$ (estimated in log space) which results in an increase in state uncertainty. This is the conceptual equivalent to forgetting. Lastly, the uncertainty of a given state is kept within realistic boundaries so that the sum of $\alpha$ and $\beta$ does not exceed 30. This is done to ensure numerical stability. See Eqs. 5–10.

*If outcome is shock ($O_t = 1$):*

$$\alpha_{(t+1,s)} = \lambda \left( \alpha_{(t,s)} + \tau^+ \right) \tag{5}$$

$$\beta_{(t+1,s)} = \lambda \beta_{(t,s)} \tag{6}$$

*If outcome is not a shock ($O_t = 0$)*

$$\beta_{(t+1,s)} = \lambda \left( \beta_{(t,s)} + \tau^- \right) \tag{7}$$

$$\alpha_{(t+1,s)} = \lambda \alpha_{(t,s)} \tag{8}$$

*Both parameters of all non-active states decay*

$$\alpha_{(t+1,s')} = \lambda \alpha_{(t,s')} \tag{9}$$

$$\beta_{(t+1,s')} = \lambda \beta_{(t,s')} \tag{10}$$

The 1-state model can behave very similarly to the more commonly used associative learning models such as the Pearce-Hall model[81,82], see Supp. Mat. For a comparison.

**Beta state inference model (n-state).** The 1-state model described above assumes a single state. However, alternatively we can let the model infer states from the data and allow for the possibility of switching between them. Specifically, our goal was for such switching model (a) to infer state switches from binary outcomes without any context cues, (b) to infer a state switch at a rate similar to humans, ideally in less than 10 trials and (c) to have a tendency to only create a handful of states to allow for meaningful generalization. The last aspect reflects the fact that every additional state (e.g., fear memory) must be maintained in parallel, but it also needs to be distinct from the already existing states. We note that a number of state switching models have been proposed previously[69,70,83,84] but none of these meets the goals set above.

As in the 1-state model, each state is characterized by a beta distribution that is updated as described above. In addition to updating each state, the model keeps track of the running average of surprise, *S* (see Eq. 11).

$$S_{(t,s)} = (1 - \pi) S_{(t-1,s)} + \pi |O_t - \widehat{P_{t,s}}| \tag{11}$$

Weighing current and past surprise using $\pi$ (Eq. 11) keeps the surprise values between 0 and 1. This, in turn, ensures that the key parameter $\eta$ is within a range that is easily interpretable.

In order to distinguish between inferring new states and state switching (it might be optimal to infer just two states but to switch between them multiple times, i.e., every time the contingencies change) the model uses two decision thresholds to guide behavior. The basic threshold is defined by the uncertainty of the current state $\sigma$ times the threshold parameter $\eta$. Exceeding this threshold triggers a polling mechanism during which all existing states are compared against an expected value which is simply the mode $\hat{P}$ of the current state +/− the current running surprise *S*. Following this procedure, the most likely next state is switched to, or the current state is kept active. If the surprise *S* exceeds the compound threshold (Eq. 13), the model first checks for any existing states in the range around the expected

value ($\pm \sigma_{(t,s)} \eta$). If multiple suitable states exist it chooses the most likely one and if none exist it creates a new state. The compound threshold is additionally controlled by the parameter *q* which represents the difficulty of creating a new state.

$$\text{(basic threshold)} \ S_{(t,s)} > \sigma_{(t,s)} \eta \tag{12}$$

$$\text{(compound threshold)} \ S_{(t,s)} > \sigma_{(t,s)} \eta q \tag{13}$$

In order to allow new states to be created but to prevent the model from creating too many states, *q* follows a Chinese Restaurant Process distribution with parameters $\theta = 0.25$ and $\alpha = 1$ under which the creation of each next state becomes progressively more difficult. CRP probability density distribution was generated using 10000 iterations of Eqs. 14 and 15 and averaging over them.

*Chinese Restaurant Process – probability of creating new state*

$$P(S_{new} = S) = \frac{\theta + |S|\alpha}{t + \theta} \tag{14}$$

*Chinese Restaurant Process – probability of choosing existing state*

$$P(S_t = S) = \frac{|s| - \alpha}{t + \theta} \tag{15}$$

When a new state is being created it is initialized with mean at the current expected value ($P_{(t,s)} \pm S_{(t,s)}$) and standard deviation calculated using Eq. 5b from the estimated parameters $\alpha_0$ and $\beta_0$ (i.e., all states will have the same starting uncertainty).

Both models were found to recover well (see the Model recovery section below for full description), providing support for a unique identifiability of the state switching strategy.

Models were also fitted to artificial data containing either one or two reinforcement levels, mimicking the stable and reversal cues from the actual task and the three contingency levels (Supp. Fig. 15). In stable environments (columns 1 and 3) the models were able to fit the data almost exactly. In environments with two reinforcement levels the 1-state model updated appropriately following contingency changes. The n-state model on the other hand was able to approximate high and low state and effectively switch between them.

## Model recovery

The 1-state and n-state models were included in a model recovery procedure. First, we fitted all models to the data of the participants. Second, we used the mean and standard deviation of the fitted parameter values to generate synthetic data. Third, the data generated using each model were fitted by each of the candidate models. The fitting procedure was identical to the one used to fit real participant data (45 runs, separate fit for each cue). Last, model comparison was performed for each artificial data set using the mean BIC as the quantitative criterion. A model was considered to recover well if the winning model matched the model used to generate the data. All investigated models recovered uniquely (see Supp. Fig. 16).

## Data quality and checks

**Shock intensity and perception.** A linear mixed effects model (LMM) was estimated to check for the differences in shock intensity between studies and its relationship with trait anxiety. The mean shock intensity did not differ between studies, $F(2,79) = 2.92$, *n.s.* nor was there a statistically significant interaction with trait anxiety $F(2,80.8) = 0.02$, *n.s.* Kolmogorov-Smirnov tests found no credible evidence for a difference in shock intensity values between full dataset and dataset after exclusions. Lastly, there was credible evidence for an association between shock intensity and reported probabilities in either low or high state indicating that probability ratings did not differ due to the participants general sensitivity to electrical stimuli. We also tested

whether the perceived shock unpleasantness and pain intensity correlated with true shock intensity or trait anxiety. If the calibration procedure had been successful, the objective shock intensity should not relate to the subjective ratings. Employing a LMM, we found no credible evidence for an association between the subjective painfulness/unpleasantness and the objective current or trait anxiety.

**Trait anxiety.** We tested whether anxiety scores differed between the three studies. While there was no statistically significant difference, $F(2,157) = 1.35$, *n.s.*, the median TA in Experiment II was 35 compared to 42 in 1 and 41 in Experiment III. We therefore decided to include experiment as a random effect in all linear mixed effect models. On occasions, anxiety results are shown as median split for convenience. Where possible, such plots are accompanied by parametric visualization. All statistics are performed using a full range of trait anxiety scores. Kolmogorov-Smirnov tests found no credible evidence for a difference in anxiety scores between the full data set and the data set after exclusions.

**Cue appeal.** Although fractals were randomly allocated to the different conditions across participants, there was a possibility that participants would rate a specific fractal more favorably due to its visual appeal. To check whether this was the case, the visual appeal ratings collected at the end of the task were included as dependent variables in a LMM with cue and session as fixed effects. LMM found no significant effect of cue or contingency on visual appeal of the presented cues.

**Initial bias.** To test whether the first rating differed from the unbiased estimate of 0.5 indicating a pre-existing bias in shock expectancy, each participant's first rating of the first session was entered into a one-way *t*-tests (separately for each experiment). These analyses did not reveal any significant effect. Since there was a degree of variability around the mean we next tested for an association between trait anxiety and the first rating, but no relationship between the variables was found.

**Session order.** In Experiment III, the three contingency conditions (i.e., sessions) were presented in a random order. To verify that our findings are not a result of an order effect, we used a LMM to test whether the session order had an influence on mean ratings separately for the high and low state of the reversal cue. The model found no significant effect of the order in which the sessions were delivered on the probability ratings.

**Starting contingency of the reversal cue.** Next, we assessed whether ratings later in the task were influenced by the starting contingency (high vs low) of the reversal cue. To perform this analysis, we removed the first half of each time course (there would of course be an effect in the early ratings, here we are checking for any lasting anchoring bias) and fitted a LMM with state, contingency, experiment and starting contingency as fixed effects. There was no significant main effect or interaction of starting contingency. By adding trait anxiety to the model, we further checked whether there was any interaction with TA but found no credible evidence for a relationship between starting contingency on trait anxiety.

### Reporting summary
Further information on research design is available in the Nature Portfolio Reporting Summary linked to this article.

## Data availability
The behavioral data generated in this study have been deposited to GitHub and are openly accessible here: https://github.com/ozika/trait-anxiety-and-state-inference-zika2023[85]. The raw behavioral data have been anonymized and are stored in a publicly available repository: https://github.com/ozika/trait-anxiety-and-state-inference-rawdata-zika2023[86].

## Code availability
The code used to derive statistical results is stored in the associated GitHub repository together with the data: https://github.com/ozika/trait-anxiety-and-state-inference-zika2023[85]. The repository includes instructions to reproduce the results, including dedicated computational virtual environment in R.

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

## Acknowledgements

We would like to thank Judith Appel and Lorika Shkreli for help with data collection for Experiment II, and to Sam Hall-McMaster and Fabian Renz for comments on the manuscript. Funding: Medical Research Council, UK (scholarship to O.Z.), The Engineering and Physical Sciences Research, Council, UK (scholarship O.Z.), NWS was funded by an Independent Max Planck Research Group grant awarded by the Max Planck Society (M.TN.A.BILD0004), the Federal Ministry of Education and Research (BMBF) and the Free and Hanseatic City of Hamburg under the Excellence Strategy of the Federal Government and the Länder and a Starting Grant from the European Union (ERC-StG-REPLAY-852669), MQ: Mental Health Research (Number MQ14F192, awarded to A.R.). K.W. was supported by an MRC UK New Investigator grant (MR/L011719/1). M.B. was supported by the Oxford Health NIHR Biomedical Research Centre and Oxford Health Clinical Research Facility. The views expressed are those of the authors and not necessarily those of the NHS, the NIHR or the Department of Health. This research was funded in whole, or in part, by the Wellcome Trust (Grant numbers 203139/Z/16/Z and 203139/A/16/Z). For the purpose of Open Access, the author has applied a CC BY public copyright licence to any Author Accepted Manuscript version arising from this submission.

## Author contributions

The following list of author contributions is based on the CRediT taxonomy. Conceptualization: O.Z., K.W., N.W.S.; Data curation: O.Z.; Formal analysis: O.Z., N.W.S.; Funding acquisition: K.W., N.W.S.; Investigation: O.Z.; Methodology: O.Z., K.W., N.W.S.; Project administration: O.Z., N.W.S.; Resources: K.W., N.W.S.; Software: O.Z., N.W.S.; Supervision: K.W., N.W.S.; Validation: O.Z.; Visualization: O.Z.; Writing - original draft: O.Z., K.W., A.R., M.B., N.W.S.; Writing - review & editing: O.Z., K.W., A.R., M.B., N.W.S.

## Funding

## Competing interests

M.B. has received travel expenses from Lundbeck for attending conferences, and has acted as a consultant for J&J, Novartis and CHDR. The remaining authors declare no competing interest.
