## [Peer Review File · Nature Communications]

Trait anxiety is associated with hidden state inference during aversive reversal learningREVIEWER COMMENTS

Reviewer #1 (Remarks to the Author):

This paper by Zika et al. represents an innovative investigation of the role of hidden state inference during aversive learning and its relationship to trait anxiety. Their results reveal that individuals with high levels of trait anxiety infer new hidden states more readily than those low in trait anxiety, and this has implications for our understanding of how anxiety manifests, and resists intervention, in the real world. The approach is novel and thorough, including some elegant computational modelling, and the results appear to be robust. I have a few comments that could be addressed in a revision, most of which are minor comments and queries:

1. In the results section, the number of participants is given across different experiments, however it reads as though three separate experiments were conducted which gives the impression the sample size was smaller than it is. It would be worth making it clear that subjects were combined across these experiments so that the full combined sample is 89 subjects.
2. In the analysis of learning after reversal (page 10), are some of the effects confounded by pre-switch probability estimates? It seems from Figure 3B that low anxiety subjects overestimate the shock probability prior to the switch, which could explain the difference in slopes post-switch?
3. It would be interesting to see how many participants were best fit by the n-state model in each condition – were there distinct groups of subjects who used hidden state inference and who didn't, and how did these differ across conditions?
4. The model fitting results are convincing, but I'm left wanting a deeper understanding of what exactly it is about the n-state model that makes it perform better in people with high levels of anxiety. This is largely because it seems that the n-state model does seem to fit best across most subjects, so presumably there is something different about the parameters of the model itself that lead to the relative improvement in model fit in highly anxious people. Would it be possible to explore this a little more to reveal where exactly the difference lies? E.g., do more anxious people have a lower threshold parameter? Presumably the 1-state model is a special case of the n-state model, in which additional state formation does not occur due to, for example, a high threshold parameter or some difference in how the average surprise is calculated, and so this single model should be able to capture the spectrum of responses through from pure gradual learning through to sensitive hidden state inference. Apologies if this is a vague question – I'm happy for the authors to address this however they see fit, or ignore this point if they feel it is not worth pursuing further.
5. Were the alpha and beta starting parameters in the model related to trait anxiety? And relatedly, were estimated levels of uncertainty in the model correlated with trait anxiety?
6. Would it be possible to infer the surprise in the model based on the beta learning rule itself, rather than introducing an additional delta learning component? E.g., based on the uncertainty in the beta distribution? The authors' approach is probably the most straightforward one, and this is just a point of curiosity rather than anything that necessarily needs to be changed.
7. For model recovery, it would be helpful to present a confusion matrix showing the proportion of correctly recovered datasets – the current figure is a little hard to interpret.
8. It might be worth mentioning in the discussion the fact that this task was completed in the scanner, and as a result may have a non-representative sample with respect to anxiety (see Charpentier et al, 2021, SCAN)
9. There are a few additional relevant papers that could be cited: Gagne et al. (2020, eLife) on links between learning in volatile environments and anxiety/depression, and Tzovara et al. (2018, PLOS CB) and Wise et al. (2019, PLOS CB) on the use of Bayesian learning models in aversive learning tasks.
10. On page 13, one result is given as $p < .05$ – could the authors provide the exact p value?

Reviewer #2 (Remarks to the Author):

Overview:

This is a behavioral study assessing if trait anxiety modulates whether individuals use gradual learning processes to update aversive learning contingencies or if they perceive latent states in the learning environment that facilitate learning. This study builds nicely on previous work in the field that tests similar predictions in fear learning paradigms (specifically, if individual differences in state inferences determine fear recovery). Healthy participants completed an aversive reversal learning task in which cues were assigned a consistently higher or lower probability of aversive shock, while another served as a reversal cue that switched between high and low shock probability. Probability ratings were reported on each trial and served as the primary DV. The authors tested if a 'gradual updating' model or 'state-inference' model provided a better account of the data. The latter was determined to provide a better fit, suggesting that variability in trait anxiety may be related to the tendency to detect distinct states in the learning environment and that this learning tendency may account for fear relapses seen in clinical samples with elevated trait anxiety. I like this study and find the question, approach and results interesting. The paper is very well-written, clear and the introduction in particular provided a nice overview of the literature and state of the field. Overall, the approach is rigorous. However, the sample sizes are quite small and call for replication in a larger sample, especially given the computational modeling approach and focus on individual differences. Other questions and comments/concerns regarding the findings are outlined below:

1. Power/replication: While I applaud the rigor of the approach and analysis, the sample sizes are currently on the smaller side ($n=30$, $n=22$, and $n=37$ for studies 1-3, respectively). Combining the data sets for the 75/25 probabilities boost the sample size considerably but I'm still concerned that these sample sizes constrain the authors ability to make strong claims about their findings. I also didn't see a power analysis in the paper so this could be added prior to the replication attempt. This is not to negate the importance of the findings, but it would make a stronger impact on the field and readers if the results were replicated in a larger sample. The authors might want to consider an online replication using an aversive outcome conducive to this means of data collection (negative feedback, small monetary loss, aversive images, etc.) instead of electric shock. In fact, it would be nice if the results generalized across domains in this way.

2. Learning differences by TA: I found it striking that the higher TA subjects appear to be more accurate in their predictions of shock probability relative to lower TA subjects (fig 2D) during the stable cue predictions. This also seemed to be the case with the reversal analyses (p10), so it doesn't appear to be unique to the stable cues. This finding counters the common assumption in the literature that more anxious individuals over- and under-estimate the likelihood of threat and safety, respectively. What do the authors make of these findings? Do they believe this is a finding specific to their task or that trait anxiety does not target expected probability of aversive outcome but some other feature of these events? How do these results compare to previous work of one of the authors (Browning et al 2015) where trait anxious individuals were unable to track volatility in aversive outcomes/learning?

3. Presentation of results: While the results are generally clear and well-described, they could be made more succinct in places. For example, in the section on learning immediately after reversal the authors can omit the slope analysis details and just state that learning evolved as expected and refer readers to the corresponding figure or SI results. Or they can do this with the steepness measure that follows as it is redundant. This will allow more focus on the TA findings throughout given this is the primary objective of the paper. They may want to do something similar with the more specific analyses at the end of the paper regarding features of the model fits.

4. Learning differences re: TA: Did high TA participants adjust their shock ratings faster for all probability conditions or just 90/10 compared to 60/40? It is unclear from the results section as currently written and should be clarified.

5. Modeling results: I had a similar question for the modeling difference results. In terms of the effect of TA on state inference, the authors report a positive association between TA and use of n-state

model on 90/10 trials, and that this association was stronger than the other probabilities (60/40, 75/25). Was this effect of TA specific to 90/10 trials? If so, it is unclear how much this actually reflects a tendency for higher TA participants to use the n-state model in general, given the 90/10 trials are also the easiest to segment into different states as opposed to updating gradually. Was this effect significant for the other probability trials? If not, it is unclear if the high TA participants are actually using a distinct learning model that is novel and speaks to something mechanistic, or if they are just more accurate overall in their predictions (as the model-free choice analysis suggests) so this biases the model-based analyses to detect a better fit for n-state learning to dominate during the 90/10 probabilities.

6. Mechanism: What do the authors propose is the mechanism behind these distinct state dependent learning strategies? Does the fact that the n-state model provided a better fit for the 90/10 probabilities suggest this form of learning might emerge only when it is optimal (i.e., cognitively easier) to segment learning easily into different states but switch to gradual learning when the probability states are more difficult to disentangle?

7. Instructions: Were the subjects made aware during the instruction period that the accuracy of their shock probability rating was independent of whether they receive a shock or not on that trial? One could imagine a scenario where participants believe the two are related and this would create an incentive to be more accurate to avoid punishment of shock. This is important to clarify to readers because it could point to a mechanism through which higher TA leads to better accuracy.

Reviewer #3 (Remarks to the Author):

In this paper, Zika and colleagues investigate the effects of trait anxiety in inferring changes in aversive environments. The authors use computational modeling to shed light on the inference process that subserves the learning of aversive environments and find that high trait anxiety is associated with more "context-specific" learning, in which more number of "states" are inferred to explain changes in the aversive environment. This paper asks questions that have clear theoretical and clinical contributions to the field. However, I have a few points that I'd like to be addressed.

1. What are the behavioral consequences of inferring distinctive states? In page 12, the authors show that those who engage in state inference show faster learning (Figure 5A). While this is a good sanity check, I'd like to see if the model parameter results stand on out-of-sample behavioral signatures. Would it be possible to fit the model to first half of the trials and see if the later half's behavioral signature are correlated with the model parameters? Additionally, were the model fit improvements with n-state model consistent across sessions within participant?

2. One of the key manipulations of this paper involves different levels of uncertainty. As the authors describe in the introduction, inferring distinctive states from stark differences (e.g., 90/10) would be easier than less obvious changes (e.g., 60/40), and the effects of trait anxiety was more pronounced in the large contingency difference condition. However, the theoretical reasoning behind this manipulation is unclear to me. Relatedly, the number of trials needed for reversal learning would be different across the probability conditions, and thus using the equal number of trials for the "meaningful" and "oddball" trial distinction does not seem appropriate. What are the differences in inference process between those meaningful and oddball trials? How are you defining "state awareness" in the inference model? For instance, do you expect the threshold to be changing as a function of trial from state transition?

3. It is interesting and somewhat counterintuitive that the effects of trait anxiety is stronger in the low state where the shock probability is overestimated in individuals with lower trait anxiety. Would the authors expect any relationship to other behavioral or physiological markers (e.g., SCR)? I am curious what would be the the implication on clinical population with regard to general vigilance. I would appreciate more discussion on this point.

4. Additional comments:

1. As far as I understand, the state inference models were fit to individual cues and there were no carry over between sessions. I would be interested in potential order effects on inference. Is it easier to deploy state-switching once you inferred that there are harmful and safe states? That is, when 90/10 session comes before 60/40 session, do you see more fit improvements for the n-state model in the 60/40 session?
2. The participants of this study are pooled from three studies, and one of the studies involved a drug administration. Although I understand that only the placebo group was included for the analysis to minimize the differences between studies, I am curious if there was any significant behavioral differences between the studies.
3. Task design: Are "sessions" and "conditions" used interchangeably? Within a session, were three cues presented in a pseudo-randomized order? It would be great if this can be clearly conveyed in Figure 1B and 1C.
4. I find reporting of the data using median split confusing. I understand the rationale to visualize the results for high and low trait anxiety participants, but the interpretation in the text makes it somewhat unclear how the results from the linear mixed models match up with the interpretation. I suggest changing the languages to reflect the statistical models used in the analyses.
5. In the computational model sections, some of the notations are missing or have typos. For example, did the authors mean $f(x,a,b)$ in Eq. 1? I believe Eq. 3 needs notations for P and O, although I can infer that they are probability and outcomes, respectively. CRP distribution with the theta and alpha parameters should be added. Could you explain how these parameter values were picked?

POINT BY POINT REPLY TO REVIEWER COMMENTS

Zika Et al. NCOMMS-22-16778A

Reviewer 1

R1-1 *In the results section, the number of participants is given across different experiments, however it reads as though three separate experiments were conducted which gives the impression the sample size was smaller than it is. It would be worth making it clear that subjects were combined across these experiments so that the full combined sample is 89 subjects.*

Thank you for your comment. We adjusted the text on pg. 5 (first paragraph of Results) as follows:

“Eighty-nine participants (44 female, mean age: 25.5 years) performed a probabilistic aversive reversal learning task during which they saw one of three possible cues and were then asked to rate the probability of receiving a shock (Fig 1a). The dataset was acquired in three separate experiments (N=30, N=22, N=37). Experiments I and II consisted of one condition (75/25, see below), Experiment III comprised three sessions, with each session differing in outcome uncertainty. Therefore, the number of participants differs between sessions (N_60/40 = 36; N_75/25 = 88; N_90/10 = 37; see Methods and Supplementary Materials for a detailed breakdown).”

R1-2 *In the analysis of learning after reversal (page 10), are some of the effects confounded by pre-switch probability estimates? It seems from Figure 3B that low anxiety subjects overestimate the shock probability prior to the switch, which could explain the difference in slopes post-switch?*

We agree that pre-switch baselines could in principle have had an influence on post-switch slopes. To test this formally, we extracted the mean ratings on 5 trials before each reversal and regressed them from the data. We then repeated the analysis reported in the main text focusing on the effect of trait anxiety over the sessions. As reported in Results, we still find the main positive effect of trait anxiety on slopes, $F(1, 146) = 8.59, p = .004$ as well as interaction between trait anxiety and session, $F(2, 542) = 5.75, p = .003$ driven by positive association between slopes and TA in the 90/10 condition, $\beta = 7.38, CI_{95} = [3.84, 10.91]$ (all $\times 10^{-4}$). The association of slope and TA in 90/10 was significantly higher compared to 60/40, $t(460) = -2.39, p = .045$, and 75/25, $t(574) = -3.31, p = .003$. We also performed this analysis on the “steepness” measure (i.e., sigmoid-based estimates of steepness), replicating the same result. We added this control analysis to the Supp. Mat., section “Control analysis of slope”.

R1-3-a *It would be interesting to see how many participants were best fit by the n-state model in each condition*

Please find the percentages of participants best fitted by the 1-/n-state models in the table and plot below. We also included this in the Modeling section (see snippet below) and the plot in Supp. Mat.

	60/40	75/25	90/10
1-state	41.7%	43.2%	37.8%
n-state	58.3%	56.8%	62.2%

From Results, p 12.:

Testing the model across all sessions, the n-state model fitted the data better (1-state BIC: -118; n-state BIC: -123). This was also true when comparing model fit for all three sessions individually, 60/40 (-84 vs -91), 75/25 (-133 vs -143) and 90/10 (-116 vs -132). However, the most pronounced difference was found in the 90/10 session where the n-state model improved fit substantially (see Fig 5b). The same pattern emerged when assessing the percentages of participants best fitted by each model: 41.7% vs 58.3% in 60/40; 43.2% vs. 56.8% in 75/25 and 37.8% vs 62.2% in 90/10 (1-state vs. n-state respectively).

R1-3-b ...were there distinct groups of subjects who used hidden state inference and who didn't, and how did these differ across conditions?

To assess this question we summarized data according to the strategies used by each participant in each of the three sessions. In the plot below, "1" stands for gradual (1-state) learning while "n" stands for structure learning (n-state). For example "1-1-n" therefore represents a case where in the 60/40 and 75/25 conditions the 1-state model fitted best and in the 90/10 condition the n-state model fitted best. Summarizing data in such a way allows us to assess strategy switches between sessions. We plot the results below.

Interestingly, there appears to be a high degree of internal consistency - bins with the same strategy across the three sessions (1-1-1, n-n-n) seem to stand out (38%). This is followed by a group of participants which relied on gradual learning in the more noisy conditions but employed structure learning in the 90/10 condition (1_n_n and 1_1_n; together 32%). We added this information to Results (see snippet below) and to Supp. Mat.

From Results, p 13.:

There was also considerable within-participant consistency of winning model across sessions. Namely, 38% participants (chance level: 11%) were fitted by the same model in all three sessions. Further 32% of participants relied on gradual learning in 60/40 but switched to state inference strategy in 90/10. For a full breakdown of within-participant strategy by session see Supp. Mat.

R1-4 *The model fitting results are convincing, but I'm left wanting a deeper understanding of what exactly it is about the n-state model that makes it perform better in people with high levels of anxiety. This is largely because it seems that the n-state model does seem to fit best across most subjects, so presumably there is something different about the parameters of the model itself that lead to the relative improvement in model fit in highly anxious people. Would it be possible to explore this a little more to reveal where exactly the difference lies? E.g., do more anxious people have a lower threshold parameter? Presumably the 1-state model is a special case of the n-state model, in which additional state formation does not occur due to, for example, a high threshold parameter or some difference in how the average surprise is calculated, and so this single model should be able to capture the spectrum of responses through from pure gradual learning through to sensitive hidden state inference. Apologies if this is a vague question – I'm happy for the authors to address this however they see fit, or ignore this point if they feel it is not worth pursuing further.*

We would like to thank the reviewer for this interesting question! We believe that this point can be split into two related questions:

1. What behavior is the n-state model preferentially fitting to?
2. Which feature/parameter of the n-state model is most important, and how does this relate to anxiety?

To answer Question 1 we considered what behavioral factors would provide evidence for state inference. One key data point where we would expect a clear delineation between gradual learning and state inference is in the difference between post-reversal learning and learning from oddball events. When a gradual learner with $\alpha=1$ experiences either a reversal or an oddball, they will learn equally from both. A state inference learner, in contrast, will react to the reversal, but dismiss the oddball. Therefore, it is important to consider steepness and oddball learning in conjunction; the steepness/slope alone may just reflect a high learning rate. We quantified steepness as the slope on trials 1-10 after true reversal (though also see Supp. Mat. for steepness estimation using the cumsum+sigmoid fit method). High TA was associated with steeper updating, particularly in the 90/10 condition ($\beta_{60/40} = 1.85 \times 10^{-4}$, $\beta_{75/25} = 1.10 \times 10^{-4}$, $\beta_{90/10} = 7.48 \times 10^{-4}$) as well as with model fit towards state inference, $r(87)=.36$, $p<.001$. We quantified learning from oddballs as the difference between learning immediately after reversal versus learning from surprising events that occurred at least 5 trials after reversal. We found that both high TA and better fit of the n-state model were associated with less learning from oddball events (see figure below). Therefore, both of these behavioral markers relate to trait anxiety and preferential fit of the n-state model. This suggests that differential behavior towards oddballs vs reversals is a key aspect that the n-state model picks up. We report the full results on pages 10, 13 and 15.

We also tried to improve the paragraph where we introduce the behavioral markers in relation to relative model fit, from p. 13:

Steeper post-reversal learning was found in participants with better relative fits of the *n*-state model (Fig 5a, see also Supp. Fig. 1). To quantify this impression, we assessed two major markers of state inference: post-reversal slope and learning from oddball events. First, we correlated the differences in model fit against the fitted slopes from participants' shock ratings (Fig. 4). This revealed a significant positive association across all three sessions, $r(87)=.36$, $p<.001$, indicating that improved fit of the *n*-state model related to the steepness of estimated switches. Second, we reasoned that those participants employing a state inference strategy should be better at dissociating when to learn from outcomes, i.e., they should show less learning from oddball events compared to learning from trials just after reversal. To test this, we calculated model-free learning rates separately for 5 trials immediately after reversal (i.e. "meaningful learning") and trials during the relatively stable periods between trial 5 and the next reversal ("oddball trials", see Methods). Participants who were fitted better by the *n*-state learned more from outcomes occurring after reversal compared to oddballs (alpha difference = 0.059) while participants fitted better by the 1-state model had a smaller difference in learning rates (alpha difference = 0.021), $t(80)=-2.20$, $p=.030$

Regarding the question which feature of the model is most important, we analyzed the internal parameters of the winning model. Interestingly, there was no relationship between any of the parameters and TA. Most notably, the Spearman correlation coefficients were between trait anxiety and the threshold η ($r=-.11$), the shock learning rate τ ($r=.09$), the no shock learning rate τ ($r=.13$), and λ ($r=.056$) all were non significant in the crucial 90/10 condition (all $ps >.45$). This strengthens our argument that the two models do not reflect faster learning or different thresholds, but a different mechanism, namely the ability to infer multiple states. We reasoned that any fit improvement over the 1-state model must be due to increased tendency towards state inference, i.e., that the differentiating factor is *the mechanism* itself.

Text from p. 12:

Note that although the *n*-state model has one additional parameter (the threshold), it can behave almost identical to the 1-state model when the threshold is so large that the model never creates more than one state. Lower BIC scores of the *n*-state can therefore be attributed to the necessity of inferring and switching states, i.e., that participants with improved model fit for the *n*-state over 1-state model are likely to rely on a state inference mechanism.

R1-5-a Were the alpha and beta starting parameters in the model related to trait anxiety?

We find this question very interesting from a theoretical standpoint: one might hypothesize that overall reduction of uncertainty is one of the mechanisms that drives state inference. To test whether the starting uncertainty estimates (alpha0 and beta0) relate to trait anxiety we constructed two LMM models. In both cases we found no main effect or interaction of trait anxiety, all LMM p-values > 0.6 (see plots below).

From p. 14:

To better understand the fitted n-state model, we explored its behavior and fitted parameters in more detail. We analyzed the internal number of states, model-estimated switchpoint, estimated uncertainty, step size parameters (τ^+ , τ^-), the threshold parameter (η), the decay parameter (λ) and the starting values of alpha and beta. These analyses found a significant effect only in the fitted step sizes for the positive and negative outcomes τ^+ and τ^- .

R1-5-b Were estimated levels of uncertainty in the model correlated with trait anxiety?

To address this point we used the standard deviation of the current state (calculated based on trial-specific alpha and beta) summed across trials as an overall estimate of uncertainty. Using a LMM, we then explored the relationship between uncertainty and anxiety. The model found no significant association between uncertainty and anxiety, or interaction with session. This is now also reported in the Supp. Mat. in the “Experienced uncertainty” section.

Albeit not significant, there was a general trend towards a negative association between TA and experienced uncertainty in the 90/10 condition as one would expect from an agent who successfully infers state structure (see plot below).

R1-6 *Would it be possible to infer the surprise in the model based on the beta learning rule itself, rather than introducing an additional delta learning component? E.g., based on the uncertainty in the beta distribution? The authors' approach is probably the most straightforward one, and this is just a point of curiosity rather than anything that necessarily needs to be changed.*

We also considered this option as it bears a certain elegance. Our approach was to use $p(O|\alpha, \beta)$ as a measure of Bayesian surprise for a given trial. Numerically, however, the *specific* values of the surprise variable either need to be calculated using normalization (when using probability density function) or arbitrary step size (when using cumulative probability density). Therefore, in order to keep the model as simple as possible, we opted for the delta rule which keeps the surprise values between 0 and 1. This, in turn, ensures that the key parameter η is within a range that is easily interpretable. We also note that our approach makes less assumptions about complex computations that would need to be carried out by our participants on the fly. Importantly, both approaches should lead to very similar results in terms of providing numerical surprise value.

R1-7 *For model recovery, it would be helpful to present a confusion matrix showing the proportion of correctly recovered datasets – the current figure is a little hard to interpret.*

Thank you for this suggestion, the confusion matrix is indeed much clearer. We have updated this in the Supp. Mat.

True model	1-state	92	8
	nstate	2	98
		1-state	nstate
		Recovered model	

R1-8 It might be worth mentioning in the discussion the fact that this task was completed in the scanner, and as a result may have a non-representative sample with respect to anxiety (see Charpentier et al, 2021, SCAN).

Thank you for bringing up this relevant point. We have added this to the discussion (see below). The mean TA is indeed slightly lower in Study 1 (40.3) compared to Study 3 (41.4) but it is actually the drug study (Study 2) which has the lowest mean trait anxiety scores (37.7).

We added the following text to Discussion (p. 17):

Additionally, previous work has identified lower TA in studies involving MRI (Charpentier et al., 2021). Therefore, the sample of study 1 (fMRI study) might not be fully representative in terms of trait anxiety. However, the mean TA level was lowest in the drug study (Study 1: 40.3; Study 2: 37.3; Study 3: 41.4) which also showed the weakest behavioral effect as reported in Supp. Mat.

R1-9 There are a few additional relevant papers that could be cited: Gagne et al. (2020, eLife) on links between learning in volatile environments and anxiety/depression, and Tzovara et al. (2018, PLOS CB) and Wise et al. (2019, PLOS CB) on the use of Bayesian learning models in aversive learning tasks.

We agree that those are relevant papers. We therefore added the following two passages:

In introduction

Building on the recent advances in computational modeling of aversive learning (Gagne et al., 2020; Tzovara et al., 2018; Wise & Dolan, 2019, 2020), we used two models that take into account each individual's precise learning history to explore the relationship between trait anxiety and state inference.

In discussion

However, in changing environments, inferring whether the objective state has changed also depends on higher order uncertainty such as volatility (Behrens et al., 2007) which has been previously associated with anxiety and depression (Browning et al., 2015; Gagne et al., 2020).

R1-10 On page 13, one result is given as $p < .05$ – could the authors provide the exact p value?

We have updated the p-value of the permutation test as follows:

From p. 13:

This result was confirmed by a permutation-tested correlation between TA and model fit improvement which was significant in the 90/10 session, $r(36) = .39$, $p = .023$ (two-tailed, corrected for multiple comparisons).

Reviewer 2

R2-1 Power/replication: While I applaud the rigor of the approach and analysis, the sample sizes are currently on the smaller side ($n=30$, $n=22$, and $n=37$ for studies 1-3, respectively). Combining the data sets for the 75/25 probabilities boost the sample size considerably but I'm still concerned that these sample sizes constrain the authors ability to make strong claims about their findings. I also didn't see a power analysis in the paper so this could be added prior to the replication attempt. This is not to negate the importance of the findings, but it would make a stronger impact on the field and readers if the results were replicated in a larger sample. The authors might want to consider an online replication using an aversive outcome conducive to this means of data collection (negative feedback, small monetary loss, aversive images, etc.) instead of electric shock. In fact, it would be nice if the results generalized across domains in this way.

We thank the reviewer for raising this important point. We have thoroughly considered the possibility of further data collection, but eventually decided against it, for the reasons described below. To summarize our arguments, we show that (a) our results already include a replication (the separate studies were intended to be just that), (b) our results are robust when scrutinized with several statistical “stress tests” (out-of-sample measures, permutation testing), (c) the power is appropriate, as indicated by a power analysis, especially considering the statistical approach taken (i.e., hierarchical modeling), and finally that (d) online experiments would differ along too many dimensions to yield easily interpretable results in the current context.

To provide more detail, we first want to note that we already took a number of steps towards replication in the original submission (something we haven't communicated clearly). Chronologically, the MRI study (Study 1) was conducted first using a state jumping model based on the Rescorla-Wagner rule (i.e., not the model used in this paper). In this study, we found the behavioral effect of anxiety and a trending relationship between state inference and TA. To follow up on this observation, we designed the three-session study (Study 3) for which we did perform a power analysis (see details below). Findings of this study replicated and extended the results of the first study. At the same time, we collected data for the drug study (Study 2). We decided to include the data from Study 2 solely to increase the statistical power. Importantly, the three studies were performed in three separate samples and are therefore independent.

Second, to follow up on the reviewer's request, we validated the key analysis (relationship between TA and relative fit of models) using a permutation test. The correlation between TA and relative model fit remained significant even when correcting for multiple comparisons and considering a two-tailed hypothesis, $r(36) = .39$, $p = .023$. We also assessed the relationship between model fit and behavioral

measures using out-of-sample analyses (a step also requested by reviewer 3). In this analysis, we find that the fit difference between the models in the *first* half of the task correlates with behavioral measures (slope, meaningful-oddball learning) in the *second* half of the task. This finding provides strong evidence that the model captures generalizable behavioral differences. For further details, please see point **R3-1-b**.

Third, the power analysis conducted prior to the start of Study 3 (in 2017) assumed a parametric relationship between trait anxiety and state inference. In order to detect a correlation of $r = 0.5$ with a power of 80% under a two-tailed hypothesis using a Bonferroni correction for multiple comparisons (corrected $\alpha = 0.0167$), a sample size of $n = 38$ participants was necessary (we also added this to the paper). Our current sample size now contains 89 participants. We recognize that some conditions have 37 data points, for that reason we also conducted a non-parametric statistical test which was previously shown to decrease Type I and II errors, and to therefore increase the analysis power (Önder, 2007). Another important consideration is that the studies are also well-powered within-subject - each participant completed more than 210 trials, and at least 6 reversals. This *is taken into account* by the linear mixed effect models that we use for analyses. Finally, we note that recent studies in Nature Comms that have used similar approaches have produced robust findings despite involving fewer participants^{1,2,3}.

Fourth, a real increase of the sample size is unfortunately not possible as we don't currently have access to the necessary in lab facilities. While the suggested online data collection would be interesting, it would involve a large number of possibly quite meaningful differences compared to the original studies - for instance, the difference between experiencing painful stimuli and receiving negative "points". It could therefore be considered a new study with its own set of hypotheses rather than a replication of the original study to increase the sample size. We fear that in the worst case, the interpretation of differences between new and original results would be inconclusive.

In sum, we found the key anxiety-inference effect in the original study which we later replicated using a well-powered independent sample. We additionally scrutinized the results using a permutation test and out-of-sample analyses. We hope that the reviewer will agree with us that the replication steps and additional analysis strongly support the robustness of the reported findings.

We also added the relevant new analyses to the manuscript.

From p. 13

Additionally, both behavioral markers of state inference (slope and meaningful-oddball learning rates) were tested using out-of-sample fits (fits from first half were related to behavioral data from second half). In both cases, the relationships remained significant. See Supp. Mat. for details.

From p. 20

Following the identification of behavioral effect in Experiment I we designed Experiment III using the appropriate power calculations. Our main effect of interest was the correlation between trait anxiety and the difference in BIC scores for the two models. To detect a 0.5 effect size at 0.8 power at least within one of the three sessions (two tailed, Bonferroni corrected $\alpha = 0.0167$) the sample of 38 participants is required. We therefore collected slightly more participants (40) in the study to meet that target.

¹ <https://www.nature.com/articles/s41467-022-33119-w> (Sept 2022)

² <https://www.nature.com/articles/s41467-022-31674-w> (July 2022)

³ <https://www.nature.com/articles/s41467-021-27618-5> (Dec 2021)

R2-2-a *Learning differences by TA: I found it striking that the higher TA subjects appear to be more accurate in their predictions of shock probability relative to lower TA subjects (fig 2D) during the stable cue predictions. This also seemed to be the case with the reversal analyses (p10), so it doesn't appear to be unique to the stable cues. This finding counters the common assumption in the literature that more anxious individuals over- and under-estimate the likelihood of threat and safety, respectively. What do the authors make of these findings?*

We thank the reviewer for bringing up this interesting point. As the reviewer correctly points out, high TA participants in the three studies were arguably *more accurate* in their shock expectancy ratings. This was most prominent in the low state which corresponds to relative safety. We agree that these results are striking - high anxiety is often associated with the inability to recognise safe environments and with the overestimation of threat. In line with this notion, a review of the literature on this topic shows fairly consistent findings in clinical samples (e.g. patients with Generalized Anxiety Disorder). However, studies using continuous trait anxiety measures in *healthy individuals* provide a less clear picture. While TA has been associated with impaired safety learning, lack of fear inhibition and increased reactivity to threat, there is also a substantial body of work either not supporting these findings (e.g., Torrents-Rodas et al., 2013; Kindt & Soeter 2014), or finding the opposite. For example, Raes et al., (2009) reported that under high cognitive load, fear extinction (indexed by SCRs) was more successful in the high TA group. In another study, Wise and Dolan (2020) found learning from safety to be *increased* in high state anxious individuals. Additionally, high trait anxiety has been associated with increased neural de-differentiation of safe and threatening contexts (see Sehlmeier et al. 2011; Indovina et al. 2011). Taken together, subclinical levels of anxiety might be beneficial to aversive learning only under certain conditions, similarly to how it is sometimes beneficial in cognitive test performance (Owens et al. 2014). This would be in line with the evolutionary conceptualization of anxiety as a state of increased vigilance to detect and avoid threats (Ledoux, 2016).

Furthermore, our task might differ considerably from those previously used in at least two key aspects: asymmetry of uncertainty between high and low states (i.e., acquisition versus extinction) and single versus repeated reversals. As Torrents-Rodas et al. (2013) suggested, the diversity of findings can partially be explained by the degree of outcome uncertainty. Studies that find large anxiety differences between acquisition and extinction also tend to use asymmetric outcome uncertainty (for example acquisition = 30% shocks; extinction = 0% shocks), see also Ojala and Bach (2020) discussing this point. The resulting differences in extinction may therefore be due to an effect of uncertainty. High uncertainty (reinforcement rate close to 50%) during acquisition might mean that individuals remain in a single subjective state, which would then manifest as resistance to extinction.

We believe that overall these mixed findings suggest that it might be useful to conduct “metaverse” analyses across the landscape of multiple parameters (in a fashion similar to Sjouwerman et al. 2022; <https://doi.org/10.1111/psyp.14130>) and to consider differences in complex computational mechanisms rather than focusing on “only” on behavioral and physiological data, just as we tried to in this paper.

Modified snippet from Discussion, p. 17-18

In our behavioral results, the effects of anxiety were driven by more accurate probability estimates in the stable-low cue and the low-state of the reversal cue, which both correspond to conditions of relative safety. This aligns with some previous reports. For example, in a gamified aversive learning paradigm Wise and Dolan (2020) reported a positive association between safety learning and state anxiety. Similarly, Raes et al., (2009) found that under a condition of higher cognitive load fear extinction (indexed by SCRs) was more successful in the high TA group. Interestingly, high anxiety (i.e., factor loading high on trait anxiety) was associated with increased engagement of cognitive control in a go/no-go paradigm (Scholz et al., unpublished), suggesting that non-clinical TA might be associated with better use of cognitive resources. The broader literature on the relationship between trait anxiety

and fear yields mixed results. While some studies report increased discrimination of CS+ and CS- (Sjouwerman et al., 2020) and comparable fear inhibition during extinction in high vs. low TA individuals (Kindt & Soeter, 2014; Torrents-Rodas et al., 2013), others report deficits in inhibitory processing (Ansari & Derakshan, 2011; Haaker et al., 2015; Myers & Davis, 2007) and safety learning (Gazendam et al., 2013; Indovina et al., 2011). Taken together, subclinical levels of anxiety might be beneficial to aversive learning only under certain conditions, similarly to how it is sometimes beneficial in cognitive test performance (Owens et al. 2014). This would be in line with the evolutionary conceptualization of anxiety as a state of increased vigilance to detect and avoid threats (Ledoux, 2016).

Modified snippet from p. 19:

Our focus was to understand how individuals with varying degrees of trait anxiety intrinsically learn and represent the structure of an aversive environment which sets the study apart from classical studies on acquisition and extinction. However, future research should systematically investigate the role of trait anxiety under different relative conditions of threat in a manner similar to Sjouwerman et al. (2022), including the difference in probabilistic versus deterministic environments.

R2-2-b *Do they believe this is a finding specific to their task or that trait anxiety does not target expected probability of aversive outcome but some other feature of these events?*

Our results suggest that trait anxiety targets expected probability - but that the strength of this effect depends on the current (perceived) context, i.e. the shock expectancy on a given trial is modulated by beliefs about the higher order structure of the task. We suggest that anxiety leads to a tendency to extract temporal patterns of the environment which, in turn, may lead to situations of fear relapse or extinction resistance (if subjective evidence for threatening state is high). Importantly, in all three studies we ensured that the *perception* of the stimuli is the same for all participants (i.e., stimulus intensity was calibrated to a pain intensity of 8/10) to rule out that differences in ratings would not be due to differences in pain intensity. Of note, the shock magnitude used in the experiments was not associated with TA.

From p. 6-7:

The calibrated stimulus intensity did not differ between studies. There was no significant relationship between shock intensity and probability ratings, or between pain intensity and trait anxiety ($p > .05$, see Methods).

R2-2-c *How do these results compare to previous work of one of the authors (Browning et al 2015) where trait anxious individuals were unable to track volatility in aversive outcomes/learning?*

We agree that this is indeed an interesting question. While we cannot provide a definite answer without running a separate study, we believe that the discrepancy in findings might be due to a number of important differences between the two studies. The task used in Browning et al. (2015) is an instrumental learning paradigm while here we used a Pavlovian task. High TA may be associated with an intact ability to track the risk of shock, but with a reduced ability to use this information appropriately when choosing between options. Another distinction is that our design did not include a stable phase. Our "stable cue" trials were always intermixed with reversal cue trials creating a generally "volatile" environment. Furthermore, participants were instructed that any cue could *change its contingency at any point*. Therefore, while the previous study focused on learning adjustments to environmental volatility *between sessions*, we report the ability of high TA to identify regularities within volatile

environments without manipulating prior experience. We added these points to discussion, see snippet below.

Alternatively, if we speculate that high TA individuals have a stronger tendency to (wrongly) infer temporal structures also in stable environments, this would result in higher learning rates and therefore smaller difference in learning rates between stable and volatile blocks as in the study by Browning et al.

From p 16:

Interestingly, high TA has previously been associated with the inability to adjust learning to environmental volatility, as reflected in a high learning rate despite stable contingencies (Browning et al., 2015). Our results show that high TA adjust their expectations faster. This might seem at odds Browning et al (2015), however, it is important to consider methodological differences, such as using instrumental, rather than Pavlovian, learning to manipulate between-session, rather than within-session, volatility.

R2-3-a *Presentation of results: While the results are generally clear and well-described, they could be made more succinct in places. For example, in the section on learning immediately after reversal the authors can omit the slope analysis details and just state that learning evolved as expected and refer readers to the corresponding figure or SI results. Or they can do this with the steepness measure that follows as it is redundant. This will allow more focus on the TA findings throughout given this is the primary objective of the paper.*

We would like to thank the reviewer for this very helpful idea. To streamline the results, we moved the section on steepness and switchpoint to the Supplement. The slope analyses indeed already demonstrated the main point that learning after reversal was faster in high TA. Additionally, we pruned Figure 4 by removing the subplots on steepness and switchpoint analyses. Lastly, we also simplified the slope analysis as follows.

From p. 10:

We next focused on the learning immediately after a reversal, i.e., trials 1 to 10 ('reversal period'). We characterized the speed of learning following a reversal by fitting a line to ratings on trials 1 to 10. This was done using a LMM with slope for each participant and state. As expected, slopes differed depending on the direction of the switch, i.e., low-to-high switches were positive (2.44%; read as 'the shock probability rating increased by 2.44% per trial') while slopes in high-to-low switches were negative (-2.33%).

Next, we took the absolute value of the slope estimates to simplify the analyses. A LMM model testing for effect of session, state and TA on slopes found a positive main effect of TA, $F(1,87)=5.85$, $p=.018$, and a main effect of session, $F(2, 96)=53.87$, $p<.001$. Post-hoc comparisons between sessions revealed that mean steepness was significantly higher for 90/10 compared to 60/40, $t_{60<90}=-9.84$, $p<.001$, and 75/25, $t_{75<90}=-7.19$, $p<.001$. Furthermore, there was a significant interaction between TA and session, $F(2, 600)=5.33$, $p=.005$. The associations were positive in all three sessions: $= 1.85 \times 10^{-4}$, $= 1.10 \times 10^{-4}$, $= 7.48 \times 10^{-4}$. Post-hoc analyses found that this was driven by a significantly stronger association between TA and slope in the 90/10 session compared to 60/40, $t_{60<90}=-2.54$, $p=.030$, and 75/25, $t_{75<90}=-3.05$, $p=.007$. Despite the overall main effect, the relationship between TA and slope in the 60/40, $F(240)=.79$, $p=.375$, and 75/25, $F(141)=.50$, $p=.482$, sessions was not significant. These results remained unchanged while controlling for the pre-reversal baseline (see Supp. Mat.).

R2-3-b They may want to do something similar with the more specific analyses at the end of the paper regarding features of the model fits.

Following this suggestion we now only report the significant effect of out step size (Tau) parameters. The remainder of the analyses of models, including the tables with parameters, was moved to the Supp. Mat..

From Results p. 14:

“We analyzed the internal number of states, model-estimated switchpoint, estimated uncertainty, step size parameters (, the threshold parameter (), the decay parameter () and the starting values of and . These analyses found a significant effect only in the fitted step sizes for the positive and negative outcomes and . A LMM with parameter type (, TA and session as fixed effects found a significant main effect of parameter type, $F(1,228)=37.03$, $p<.001$, which reflected that shocks elicited larger updates than no-shocks $=1.13$ vs. $=0.73$). There was no main effect of TA, $F(1,83)=.21$, $p=.77$, or interaction of outcome type (shock/no-shock) and trait anxiety, $F(1, 228)=.16$, $p=.65$. Note that the same two parameters of the 1-state model, had a similar difference $=1.17$ vs. $=0.86$), suggesting that differential learning from shock and no-shock events alone was unable to explain our behavioral effects of TA (see Supp. Mat. for full analyses of model parameters).”

R2-4 Learning differences re: TA: Did high TA participants adjust their shock ratings faster for all probability conditions or just 90/10 compared to 60/40? It is unclear from the results section as currently written and should be clarified.

We agree that the clarity of this section needs to be improved, and have therefore adapted the relevant section in the manuscript, see below. We believe that the lack of clarity was partly due to the fact that we analyzed the signed slope which complicates the results due to the sign flip between the high and low state reversals. To simplify the results, we re-analysed the data using absolute slope which provides a more straightforward result. To answer the reviewers question, we find that the relationship between TA and slope in the 60/40 and 75/25 conditions was not significant when considered in isolation.

The revised excerpt can be found in point **R2-3-a** (to avoid repetition).

R2-5-a Modeling results: I had a similar question for the modeling difference results. In terms of the effect of TA on state inference, the authors report a positive association between TA and use of n-

state model on 90/10 trials, and that this association was stronger than the other probabilities (60/40, 75/25). Was this effect of TA specific to 90/10 trials?

Yes, a significant relationship between TA and relative model fit was only present in the 90/10 condition. The estimated betas for the `deltafit ~ TA` effect for each of the three conditions were: -0.04, -0.07 and 0.92 (60/40, 75/25 and 90/10). We included this to the manuscript (page 13), see excerpt below:

We next examined the relationship between trait anxiety (TA) and state inference by constructing a LMM with model fit difference as the dependent variable and TA and session as fixed effects. This model identified a significant interaction between TA and session, $F(2,105)=5.20$, $p=.007$. Post-hoc analyses revealed that this was driven by a positive association between TA and fit improvement in the 90/10 session, $F(1, 153)=9.61$, $p=.002$ (see Fig. 5c). There was no significant association in the 60/40 or 75/25 sessions. Betas for the TA effect by condition were 0.04, -0.07 and 0.92 (60/40, 75/25 and 90/10). The association was significantly stronger in the 90/10 session compared to both 60/40, $t(78)=-2.64$, $p=.027$ and 75/25, $t(111)=-2.96$, $p=.011$.

R2-5-b *If so, it is unclear how much this actually reflect a tendency for higher TA participants to use the n-state model in general, given the 90/10 trials are also the easiest to segment into different states as opposed to updating gradually. Was this effect significant for the other probability trials? If not, it is unclear if the high TA participants are actually using a distinct learning model that is novel and speaks to something mechanistic, or if they are just more accurate overall in their predictions (as the model-free choice analysis suggests) so this biases the model-based analyses to detect a better fit for n-state learning to dominate during the 90/10 probabilities.*

This is indeed a very relevant question. The key aspect that distinguishes the n-state model from the 1 state model is not (only) how accurate participants are, but how they behave following reversals relative to oddballs. Highly accurate choices in the 90/10 condition can in principle be captured by the 1-state model with a high learning rate. But this model would suggest a strong reactivity towards oddball trials for the same participant. But, as we show in Fig. 6 (replicated to the right), high TA individuals tend to react less to oddballs. This speaks against a simple tendency for higher accuracy in high TA

individuals, and suggests that state information is being incorporated into the updating of current

estimates.

This of course still leaves us with the question why high TA individuals are more likely to adopt an n-state model in the 90/10 condition but not in the more noisy conditions. We can only speculate, but one possibility is that this might be related to the optimality of using a 1-state gradual learning algorithm in the conditions with higher outcome uncertainty. Trying to infer states in noisy environments can require substantial cognitive load and it might not yield much benefit. Theoretically, there should perhaps exist an optimal ratio between cognitive load and prediction accuracy. This would then drive whether it's optimal to update gradually or whether to allocate resources to identify structure. Determining the optimality of a given strategy and linking it to our findings would be a great future avenue. Specifically, investigating whether sub-clinically anxious individuals find structures in environments where it's possible (as opposed to noisy environments where meaningful versus random events are effectively inseparable) while clinically anxious populations tend to search for structures even in high noise environments would be a very valuable avenue. Part of this work should focus on examining how our state inference findings relate to the previously reported relationship between anxiety and intolerance of uncertainty, especially in relation to specific types of uncertainty (e.g. Piray and Daw, 2021).

From p. 18:

A major question that remains to be answered is why do high TA individuals rely on state inference mostly in the 90/10 condition. One possibility might be that while the uncertainty in noisy environments is too high and learning meaningful changes from stochastic events poses high cognitive demands, in 90/10, learning the structure of the environment can meaningfully result in reduction of internal uncertainty.

R2-6 Mechanism: What do the authors propose is the mechanism behind these distinct state dependent learning strategies? Does the fact that the n-state model provided a better fit for the 90/10 probabilities suggest this form of learning might emerge only when it is optimal (i.e., cognitively

easier) to segment learning easily into different states but switch to gradual learning when the probability states are more difficult to disentangle?

Our interpretation of the data is precisely what the reviewer is suggesting. Separating the environment into latent states arguably comes at higher cognitive cost which might only be justified (and relatively low) in environments with low outcome uncertainty. In simple one-dimensional environments such as in our task, optimality is likely driven by the combination of stochasticity and volatility (randomness versus true changes). In general, when stochasticity is high (as in 60/40 and 75/25) it is arguably much harder to identify true changes in the environment. Furthermore, trying to infer temporal structure in noisy environments requires integration over longer time periods which means that cognitive demands remain high for longer. See snippet that we added to discussion.

From Discussion p. 17:

Our results indicate that the propensity for state-dependent learning might depend on the amount of outcome uncertainty in the environment, since better fits of the n-state model were observed in sessions with more distinct high- and low-probability states (90% and 10%), as compared to sessions with less distinct states. An above-chance proportion of participants (34%) switched from using gradual strategy in 60/40 to using state inference in 90/10. The optimal strategy for a given environment might depend on the tradeoff between cognitive effort and accuracy in prediction. Separating the environment into latent states arguably comes at higher cognitive cost which might only be justified in environments with low outcome uncertainty. Trying to infer states in noisy environments can require substantial cognitive load (e.g., integration over longer periods of time) and it might not yield much benefit (i.e., predictive accuracy).

R2-7 *Instructions: Were the subjects made aware during the instruction period that the accuracy of their shock probability rating was independent of whether they receive a shock or not on that trial? One could imagine a scenario where participants believe the two are related and this would create an incentive to be more accurate to avoid punishment of shock. This is important to clarify to readers because it could point to a mechanism through which higher TA leads to better accuracy.*

We agree that this could indeed have influenced participants' behavior. We explicitly instructed participants that their ratings would not influence whether a shock would be delivered or not. We have clarified this in the manuscript and write on page 23:

Participants were presented with minimal information regarding the number of cues, task duration, cue frequency and switches. They were told that 'each cue is associated with a certain probability of receiving a painful stimulus' and to 'pay attention to all three cues as any of them may or may not change their probability signaling the painful stimulation at any point'. Participants were also explicitly told that their ratings do not impact the outcomes.

Reviewer 3

R3-1-a *What are the behavioral consequences of inferring distinctive states?*

Thank you for this interesting question. In short, steepness and decreased learning from oddballs. We note that steepness on its own does not *warrant* state inference, although it is a necessary requirement as switches between states are by definition abrupt. To further investigate the behavioral consequences of state inference, we explored how participants reacted to oddball events, relative to how they reacted to trials after reversal. We show that participants whose behavior was better fitted by the n-state model (and high TA) reacted less from surprising events on trials in relatively stable periods compared to trials following reversals. This indicates that they are incorporating knowledge of latent states to their learning.

Following a similar question from reviewer 1, we also tried to improve the paragraph where we introduce the behavioral markers in relation to relative model fit (see snippet below).

From p. 13:

Steeper post-reversal learning was found in participants with better relative fits of the n-state model (Fig 5a, see also Supp. Fig. 1). To quantify this impression, we assessed two major markers of state inference: post-reversal slope and learning from oddball events. First, we correlated the differences in model fit against the fitted slopes from participants' shock ratings (Fig. 4). This revealed a significant positive association across all three sessions, $r(87)=.36$, $p<.001$, indicating that improved fit of the n-state model related to the steepness of estimated switches. Second, we reasoned that those participants employing a state inference strategy should be better at dissociating when to learn from outcomes, i.e., they should show less learning from oddball events compared to learning from trials just after reversal. To test this, we calculated model-free learning rates separately for 5 trials immediately after reversal (i.e. "meaningful learning") and trials during the relatively stable periods between trial 5 and the next reversal ("oddball trials", see Methods). Participants who were fitted better by the n-state learned more from outcomes occurring after reversal compared to oddballs (alpha difference = 0.059) while participants fitted better by the 1-state model had a smaller difference in learning rates (alpha difference = 0.021), $t(80)=-2.20$, $p=.030$

R3-1-b *In page 12, the authors show that those who engage in state inference show faster learning (Figure 5A). While this is a good sanity check, I'd like to see if the model parameter results stand on out-of-sample behavioral signatures. Would it be possible to fit the model to first half of the trials and see if the later half's behavioral signature are correlated with the model parameters?*

Thank you for this point, it's indeed a great way to validate the results. We re-fitted the models to reversal phases 1 - 3 (i.e., "first half") and calculated the information criteria, taking their difference identically to the main analysis. Next, we calculated slopes and meaningful-oddball learning rate difference for phases 4+ (i.e., "second half"). In both cases, the behavioral marker from the second half correlated with the first half relative model fit. Specifically, the Spearman's rho correlation coefficients were $r=.332$, $p=.002$ for slopes and $r=.266$, $p=.013$ for meaningful/oddball. We added this to the main text (excerpt below) and the Supp. Mat. Please also see the relevant plots below.

From p. 13.:

Additionally, both behavioral markers of state inference (slope and meaningful-oddball learning rates) were tested using out-of-sample fits (fits from first half were related to behavioral data from second half). In both cases, the relationships remained significant. See Supp. Mat. for details.

R3-1-c Additionally, were the model fit improvements with n-state model consistent across sessions within participant?

In response to the reviewer's question, we correlated the relative model fit for participants who completed all three sessions (correlation matrix below). This analysis identified within-participant consistency of relative n-1-state model fit, particularly between 75/25 and 90/10 ($r=.44, p=.007$), and 60/40 and 75/25 ($r=.38, p=.023$). For further details, we would like to refer the reviewer to our reply to point R1-3-b which confirms a degree of internal consistency. We added this analysis to the Supp. Mat. ("Best model fit by condition" section)

R3-2-a *One of the key manipulations of this paper involves different levels of uncertainty. As the authors describe in the introduction, inferring distinctive states from stark differences (e.g., 90/10) would be easier than less obvious changes (e.g., 60/40), and the effects of trait anxiety was more pronounced in the large contingency difference condition. However, the theoretical reasoning behind this manipulation is unclear to me.*

We agree that this should have been explained more clearly. As correctly pointed out, the three conditions varied in the amount of outcome uncertainty. States are therefore easier to infer in environments with lower (90/10) compared to high (60/40) uncertainty, and we stipulate that it therefore is more appropriate (or optimal) to use an n-state model in the 90/10 condition than in the 60/40 condition. In other words, extracting temporally extended patterns and organizing them into structure will lead to many errors in 60/40, while cognitively simpler recency-weighted updating will do well in terms of prediction (without posing excessive burden on memory, attention, etc). We also point the reviewer to a related response to reviewer 2's point 5 (**R2-5**).

To clarify this point, we also added the following text to the Discussion (pages 17 and 18):

An above-chance proportion of participants (34%) switched from using gradual strategy in 60/40 to using state inference in 90/10. The optimal strategy for a given environment might depend on the tradeoff between cognitive effort and accuracy in prediction. Separating the environment into latent states arguably comes at higher cognitive cost which might only be justified in environments with low outcome uncertainty. Trying to infer states in noisy environments can require substantial cognitive load (e.g., integration over longer periods of time) and it might not yield much benefit (i.e., predictive accuracy).

A major question that remains to be answered is why do high TA individuals rely on state inference mostly in the 90/10 condition. One possibility might be that while the uncertainty in the noisy environments is too high and learning meaningful changes from stochastic events poses high cognitive demands, in 90/10, learning the structure of the environment can meaningfully result in reduction of internal uncertainty. An interesting future direction would be to investigate whether clinically anxious individuals continue to (perhaps sub-optimally) try and find structures in noisy environments, i.e. whether they tend to erroneously find too many latent causes, or whether they are instead driven by the adversity of high uncertainty itself and lump all experiences under a single latent cause (as in Norbury et al. 2021).

R3-2-b *Relatedly, the number of trials needed for reversal learning would be different across the probability conditions, and thus using the equal number of trials for the "meaningful" and "oddball" trial distinction does not seem appropriate. What are the differences in inference process between those meaningful and oddball trials?*

We agree that what constitutes an oddball in the 60/40 condition might be different from an oddball in the 90/10 condition. The rationale for using data after the fifth trial following reversal was based on the upper plot shown below. On the y-axis we plot the change in rating between two consecutive trials. On the x-axis we show the trial number before and after true reversal. Any values higher than 0 suggest that on average participants were updating their beliefs. It appears that most updating occurred within the first 5-7 trials. To assess the impact of the selection window more systematically, we used alternative decision criteria (i.e., cut-offs at trials 7, 10 and 13) to generate meaningful/oddball trials (see the calculated learning rates for the four analyses plotted separately for each condition below). Notably, the different cutoffs produced very similar results. They all clearly capture the main effect difference

between meaningful and oddball learning. Indeed, using a formal LMM, there was no significant main effect or interaction involving the cutoff, all $ps > 0.9$.

We also added the following text to Methods, p. 26:

To check that our specific choice of post-reversal cutoff trial (ct=5) did not drive the results, we calculated oddball/meaningful learning rates for three additional cutoff values: 7, 10 and 13. We next tested the impact of the cutoff threshold on the estimated meaningful/cutoff values using a LMM. We found no significant impact of the cutoff, all $ps > 0.9$. We also present the result in Supp. Figure 2.

R3-2-c How are you defining “state awareness” in the inference model? For instance, do you expect the threshold to be changing as a function of trial from state transition?

We indeed suggest in the discussion that the difference between learning from surprising events after reversal versus later in the stable period might reflect ‘state awareness’. To clarify, we mean that reduced learning from oddball events suggests that participants are *aware* that a single event might not signal state change. In contrast, when multiple surprising outcomes occur this likely reflects true environmental change, so state switch is in order. We clarified this in Discussion.

From Discussion p. 18:

We argue that a bigger difference in learning rates for meaningful and oddball events reflects state awareness, that is, ignoring oddball events suggests knowledge of a higher-order structure.

R3-3-a *It is interesting and somewhat counterintuitive that the effects of trait anxiety is stronger in the low state where the shock probability is overestimated in individuals with lower trait anxiety. Would the authors expect any relationship to other behavioral or physiological markers (e.g., SCR)?*

This is indeed an interesting question that should be addressed in future studies. On one hand, physiological markers could either mirror expectancy ratings and thereby reflect the individual's cognitive model of the environment. Indeed, some studies have reported alignment between self-reported and physiological measures (Michael, 2007; Torrents-Rodas et al. 2013; Tinoco-Gonzales et al. 2015). On the other hand, expectancy ratings could reflect deliberate cognitive processes whereas the physiological fear response might diverge. Divergence between ratings and physiological markers has been reported by a large number of studies (Andreatta and Pauli, 2017; Boddez et al. 2012; Kindt and Soeter, 2014; Gazendam et al. 2013; Andreatta et al. 2020). For example, Homan et al (2021) reported change in physiological responses following a reversal without contingency awareness. Speculatively, explicit ratings in anxious populations might also reflect a safety mechanism in itself. In our case, high TA participants might be aware of relative safety but not be able to inhibit fear response. In fact, Gershman and Hartley (2015) did not find a relationship between trait anxiety and SCR-indexed spontaneous recovery data.

We added the following paragraph to discussion:

One question that remains to be answered is whether the ratings-based results reported here would be followed by physiological measures. Physiological markers could mirror expectancy ratings and thereby reflect the individual's cognitive model of the environment (as e.g. in Michael, 2007; Torrents-Rodas et al. 2013; Tinoco-Gonzales et al. 2015) or they could reflect deliberate cognitive processes whereas the physiological fear response might diverge (as in Andreatta and Pauli, 2017; Boddez et al. 2012; Kindt and Soeter, 2014; Gazendam et al. 2013; Andreatta et al. 2020). For example, Homan et al (2021) reported change in physiological responses following a reversal without contingency awareness. In our case, high TA participants might be aware of relative safety (i.e. be in a low state) but not be able to inhibit fear response. In support of this idea Gershman and Hartley (2015) looked at the relationship between SCR-indexed spontaneous recovery and state inference. In one of their analyses, they report that trait anxiety was not associated with SCR-indexed inference of multiple states.

R3-3-b *I am curious what would be the implication on clinical population with regard to general vigilance.*

We thank the reviewer for this question. We suggest that at the core of this point lies the question whether the increased tendency to infer temporal structure actually exists in clinical groups. While reports on the role of TA in aversive learning in a healthy population often bear conflicting evidence, the support for a disruptive role of clinical anxiety in fear learning is fairly consistent (Duits et al. 2014), i.e., clinical populations are linked to lack of fear extinction. This would be more in line with one-state conceptualization of the environment. Indeed, early work investigating the role of state inference in PTSD patients using the classical fear extinction paradigm found that the lack of extinction is related to the patient group inferring a single latent state (Norbury et al 2021).

In relation to vigilance, one option is that non-clinical levels of trait anxiety are associated with adaptive responding while the same mechanisms become maladaptive in clinical anxiety. Under this assumption, healthy controls high in TA might be able to utilize their vigilance to infer multiple (correct)

latent states while clinical levels of anxiety might lead to over-generalization, i.e., classifying all events as having single latent cause. Alternatively, clinically anxious individuals driven by the motivation to reduce uncertainty may find *too many* latent causes (i.e., non-existent patterns in data). The latter could for example be driven by mis-attribution of structure (i.e., reducible uncertainty) to random events (i.e., stochasticity). Future work should consider these two options, as well as the relationship between state inference and vigilance-avoidance, which might compete for explanation of fear relapse (Weinberg and Hajcak, 2010).

We added a section discussing the possible relationships between state inference and clinical anxiety to Discussion. Due to space restrictions, we kept this part brief, however, if the reviewer wishes we could work in more on the role of vigilance into discussion.

Text added to Discussion p. 18:

An interesting future direction would be to investigate whether clinically anxious individuals continue to (perhaps sub-optimally) try and find structures in noisy environments, i.e. whether they tend to erroneously find too many latent causes, or whether they are instead driven by the adversity of high uncertainty itself and lump all experiences under a single latent cause (as in Norbury et al. 2021).

Additional comments - R3

R3-4-a *As far as I understand, the state inference models were fit to individual cues and there were no carry over between sessions. I would be interested in potential order effects on inference. Is it easier to deploy state-switching once you inferred that there are harmful and safe states? That is, when 90/10 session comes before 60/40 session, do you see more fit improvements for the n-state model in the 60/40 session?*

This is a great point! Using only participants who completed all three sessions, we first split the data by session (60/40, 75/25, 90/10) and the order position in which they appeared (first/second/third). Testing for any order effects using a LMM revealed no significant order effects.

Zooming in on the specific question posed by the reviewer, we focused on model fit in the 60/40 condition depending on whether participants completed this condition first or after the 90/10 condition. The mean reliance on state inference was higher when 90/10 preceded 60/40 (deltaBIC = 16.4) compared to when 60/40 occurred first (deltaBIC = 6.86), but this effect was not statistically significant, $t(42)=1.61$, $p=.11$ (left panel below). We also analyzed the relationship between TA and model fit as a function of order. We also found no significant effect here (right panel below).

We included these analyses in the Supp. Mat. ("Order effects").

R3-4-b The participants of this study are pooled from three studies, and one of the studies involved a drug administration. Although I understand that only the placebo group was included for the analysis to minimize the differences between studies, I am curious if there was any significant behavioral differences between the studies.

To compare differences between studies we re-ran the behavioral analyses (ratings, slope) using the 75/25 condition with study as a fixed effect. The LMM models did not find any significant difference between studies in either analysis. As outlined in response R1-8, the trait anxiety scores were generally lowest in the drug study (e.g., using *overall* median split across all three studies, there were only six participants in the high TA “group”). Here, we show the behavioral ratings (trials 10+) separately for each study, state and median-split anxiety. We included this analysis in the Supp. Mat. (“Differences between studies”).

R3-4-c Task design: Are “sessions” and “conditions” used interchangeably? Within a session, were three cues presented in a pseudo-randomized order? It would be great if this can be clearly conveyed in Figure 1B and 1C.

We thank the reviewer for spotting this inconsistency. The two terms were indeed used interchangeably. We now unified them under the name "session". We also adjusted Figure 1 to highlight the pseudo-randomized order of cues across trials.

R3-4-d I find reporting of the data using median split confusing. I understand the rationale to visualize the results for high and low trait anxiety participants, but the interpretation in the text makes it somewhat unclear how the results from the linear mixed models match up with the interpretation. I suggest changing the languages to reflect the statistical models used in the analyses.

We felt the same way, so in response to the reviewer’s point, we modified the manuscript in two ways. First, as suggested, we added the corresponding parametric slopes where relevant (i.e., analyses with TA). After consideration, we also kept the mean estimates for the following reason: in addition to

knowing the change in ratings as a function of TA, the readers might also be interested in knowing the approximate absolute values, for example to compare the ratings to true estimates. We are happy to modify this further if the reviewer thinks it'd be helpful. Below is a modified excerpt from the section in stable cues. The modifications can be found in the Results section (pages 8 and 10).

*This analysis revealed that the difference in ratings between high- and low-prob cues increased as a function of trait anxiety, as indicated by an interaction of TA and cue type, $F(1,308)=6.91, p=.009$, see Fig 2c. **There was a positive association with TA in stable-high cue, $=.0024$, and a negative relationship in the stable-low cue, $=-.0024$: high TA participants reported higher ratings in stable-high and lower ratings in stable-low cue.** Direct contrast of the associations of TA and rating between high and low-prob cues showed significant difference, $t(242)=2.63, p=.009$. We also tested whether ratings differed significantly from the true contingency level using one-way t -tests, see Fig 2d. When judging the stable-high cue, less anxious participants significantly underpredicted the true reinforcement level in the 75/25, $t(47)=-2.62, p=.047$, and 90/10, $t(18)=-3.51, p=.015$, sessions. When judging the stable-low cue, less anxious participants overpredicted the probability in the 75/25 session, $t(47)=3.58, p=.010$. More anxious participants, in contrast, did not show over- or underpredictions, all $p > .05$.*

Second, to help with relating the statistical models to the visualizations, we added the marginal means predicted by the corresponding linear model to each plot. This hopefully helps to match the model results to the visualizations. Below we show the revised Figure 2a with the added means as an example.

R3-4-e In the computational model sections, some of the notations are missing or have typos. For example, did the authors mean $f(x,a,b)$ in Eq. 1? I believe Eq. 3 needs notations for P and O , although I can infer that they are probability and outcomes, respectively. CRP distribution with the theta and alpha parameters should be added. Could you explain how these parameter values were picked?

We thank the reviewer for spotting these errors which have now been fixed. We also added the relevant equations and description for the CRP. The specific values were chosen to reflect a plausible

prior for the number of states with one or two states being fairly likely but 10+ states being unlikely given the task (see image of the distribution below). This prior also reflects latent causes inferred by Norbury et al (2021), Figure 2c. We decided to fix these parameters because the estimation of both the eta and the CRP parameters would have led to identifiability issues, since they impose opposing effects on the overall threshold.

In order to allow new states to be created but to prevent the model from creating too many states, q follows a Chinese Restaurant Process distribution with parameters $\alpha=0.25$ and $\beta=1$ under which the creation of each next state becomes progressively more difficult. CRP probability density distribution was generated using 10000 iterations of Eqs. 14 and 15 and averaging over them.

The following text was added to Method p. 28.

Chinese Restaurant Process – probability of creating new state

$$\text{Eq. 14 } P(S_{\text{new}}=S) = \frac{\theta}{\theta + |S|}$$

Chinese Restaurant Process – probability of choosing existing state

$$\text{Eq. 15 } P(S_t=S) = \frac{s}{s + \alpha}$$

When a new state is being created it is initialized with mean at the current expected value (P_t, sSt, s) and standard deviation calculated using Eq. 5b from the estimated parameters θ and α (i.e., all states will have the same starting uncertainty).

Additional changes

ADD-1

During correction of the plots we noticed that one of the scripts had accidentally excluded 18 out of 89 participants from the analysis of the ratings for the reversal cue (Page 11). This did not qualitatively impact the results, but some of the statistical values have changed slightly.

REVIEWERS' COMMENTS

Reviewer #1 (Remarks to the Author):

The authors have addressed my comments very thoroughly and I am happy to recommend publication.

Reviewer #2 (Remarks to the Author):

The authors have done a nice job responding to my previous comments and concerns. I still believe a larger replication sample would have been ideal, especially given the strongest effect was seen in the 90/10 uncertainty condition and this was only made up of 37 participants. Otherwise, I liked the additional analyses and feel the results are much clearer now.

Reviewer #3 (Remarks to the Author):

The authors have well addressed the major issues in their revised manuscript. I thank the authors for their efforts and congratulate them on this work.

Here are a few minor points that remain:

- Figure 1. (b) caption has a typo (sesions → sessions)
- Eq. 2 (previously Eq. 3) still seems to need notations for P, O, and t (see R3-4-e)
- Page 26. The newly added description has a typo (Figure X)
- The first paragraph of the new discussion included in R3-2-a does not seem to be in the main text. It would be fine to have this just in the response as the second paragraph does explain the reasoning well, but I am just bringing this to the authors' attention.

Response to reviewers

Reviewer #1

R1.1: *The authors have addressed my comments very thoroughly and I am happy to recommend publication.*

We thank the reviewer for their helpful and positive feedback.

Reviewer #2

R2.1: *The authors have done a nice job responding to my previous comments and concerns. I still believe a larger replication sample would have been ideal, especially given the strongest effect was seen in the 90/10 uncertainty condition and this was only made up of 37 participants. Otherwise, I liked the additional analyses and feel the results are much clearer now.*

We thank the reviewer for their helpful and positive feedback. We agree with the general sentiment and hope we will be able to continue and replicate the finding in a new experiment with a 90/10 condition.

Reviewer #3

R3.1: *The authors have well addressed the major issues in their revised manuscript. I thank the authors for their efforts and congratulate them on this work.*

We thank the reviewer for their helpful and positive feedback.

R3.2: *Here are a few minor points that remain: Figure 1. (b) caption has a typo (sessions ??? sessions)*

The typo has now been fixed.

R3.3: *Eq. 2 (previously Eq. 3) still seems to need notations for P, O, and t (see R3-4-e)*

We thank the reviewer for noticing this. We have now amended the text as follows:

"To obtain trial-wise learning rates, we rearranged the Rescorla-Wagner (Eq. 1) learning rule and calculated the trial-specific learning rate α (Eq. 2), where P stands for probability ratings, O for outcomes (shock/no-shock) and t for a given trial. "

R3.4: *Page 26. The newly added description has a typo (Figure X)*

We have now fixed this to read: "Supplementary Figure 3."

R3.5: *The first paragraph of the new discussion included in R3-2-a does not seem to be in the main text. It would be fine to have this just in the response as the second paragraph does explain the reasoning well, but I am just bringing this to the authors' attention*

We thank the reviewer for bringing this to our attention. We actually found the first paragraph to raise important points that were not covered in the rest of the discussion, so we now included it.